# Atmospheric chemical loss processes of isocyanic acid (HNCO): a combined theoretical kinetic and global modelling study

Simon Rosanka[1], Giang H. T. Vu[2], Hue M. T. Nguyen[2], Tien V. Pham[3], Umar Javed[1], Domenico Taraborrelli[1], Luc Vereecken[1]

[1] Institute for energy and climate research, Forschungszentrum Jülich Gmbh, Jülich, Germany

[2] Faculty of Chemistry and Centre for Computational Science, Hanoi National University of Education, Hanoi, Vietnam

[3] School of Chemical Engineering, Hanoi University of Science and Technology, Hanoi, Vietnam

Correspondence to: Hue M.T. Nguyen (hue.nguyen@hnue.edu.vn) or Domenico Taraborrelli (d.taraborrelli@fz-juelich.de)

**Abstract**

Isocyanic acid (HNCO) is a chemical constituent known to be harmful to humans if ambient concentrations exceed ~1 ppbv. HNCO is mainly emitted by combustion processes, but is also inadvertently released by $NO_x$ mitigation measures in flue gas treatments. With increasing biomass burning and more widespread usage of catalytic converters in car engines, good prediction of HNCO atmospheric levels with global models is desirable. Little is known directly on the chemical loss processes of HNCO, which limits the implementation in global Earth system models. This study aims to close this knowledge gap by combining a theoretical kinetic study on the major oxidants reacting with HNCO with a global modelling study. The potential energy surfaces of the reactions of HNCO with OH and $NO_3$ radicals, Cl atoms, and ozone, were studied using high-level CCSD(T)/CBS(DTQ)//M06-2X/aug-cc-pVTZ quantum chemical methodologies, followed by TST theoretical kinetic predictions of the rate coefficients at temperatures of 200-3000K. It was found that the reactions are all slow in atmospheric conditions, with k(300K) $\leq$ $7\times10^{-16}$ $cm^3$ molecule$^{-1}$ s$^{-1}$, and that product formation occurs predominantly by H-abstraction; the predictions are in good agreement with earlier experimental work, where available. The reverse reactions of NCO radicals with $H_2O$, $HNO_3$, and HCl, of importance mostly in combustion, were also examined briefly.

The findings are implemented into the atmospheric model EMAC to estimate the importance of each chemical loss processes, on a global scale. The EMAC predictions confirm that the gas phase chemical loss of HNCO is a negligible process, contributing less than 1%, leaving heterogeneous losses as the major sinks. The removal of HNCO by clouds and precipitation contributes for about 10% of the total loss, while globally dry deposition is the main sink, accounting for ~90%. The global simulation also shows that due to its long chemical lifetime in the free troposphere, HNCO can be efficiently transported into the UTLS by deep convection events. Daily average mixing ratios of ground level HNCO are found to regularly exceed 1 ppbv, in regions dominated by biomass burning events, but rarely exceed levels above 10 ppt, though locally instantaneous toxic levels are expected.

**1 Introduction**

The existence of isocyanic acid (HNCO) in the atmosphere has been established only recently (Roberts et al., 2011; Wentzell et al., 2013) despite its molecular structure and chemical synthesis being first discovered in the 19[th] century (Liebig and Wöhler, 1830). HNCO can form H-bonded clusters (Zabardasti et al., 2009, 2010; Zabardasti and Solimannejad, 2007), and in pure form appreciably polymerizes to other species (Belson and Strachan, 1982), but becomes relatively stable in the gaseous phase (ppm level) under ambient temperature conditions (Roberts et al., 2010). It is thus near-exclusively present as a monomer in the gaseous phase under ambient temperature conditions (Fischer et al., 2002; Roberts et al., 2010). The background ambient mixing ratios of HNCO as determined by Young et al. (2012) using a global chemistry transport model vary in the range of a few pptv over the ocean and remote Southern Hemisphere, to tens of pptv over landmass. In urban regions, HNCO mixing ratio increases from tens of pptv to hundreds of pptv (Roberts et al., 2014; Wentzell et al., 2013). Peak levels can reach up to a few ppbv under the conditions impacted by direct emissions (Chandra and Sinha, 2016).

HNCO has been linked to adverse health effects such as cataracts, cardiovascular disease, and rheumatoid arthritis via a process called protein carbamylation [see (Leslie et al., 2019; Roberts et al., 2011; Suarez-Bertoa and Astorga, 2016; SUVA, 2016; Wang et al., 2007) and references therein]. To our knowledge, no past studies have been performed to provide a direct link between inhalation exposure and related adverse health effects. However, human exposure to HNCO concentrations of 1 ppbv is estimated to be potentially sufficient to start the process of protein carbamylation (Roberts et al., 2011). Unfortunately, an air quality standard for HNCO does not exist in most of the countries, whereas an occupational exposure limit has been established by law in only a few countries, including the Swedish Work Environment Authority (SWEA, 2011) and the Swiss National Accident Insurance Fund (SUVA, 2016). For example, the Swedish work environment authority sets the level limit value (LLV) for HNCO at about 0.018 mg m$^{-3}$, i.e. 10 ppbv (SWEA, 2011). The potential negative impact on health makes it important to assess the atmospheric sources and sinks of HNCO to determine its fate and lifetime.

HNCO emission into the atmosphere is driven primarily by combustion processes based on both natural and anthropogenic activities [see (Leslie et al., 2019) and references therein], where the pyrolysis of nitrogen-containing biomass materials during the events of wildfires and agricultural fires leads to the emission of HNCO into the atmosphere. The presence of HNCO in cigarette smoke has been established via the pyrolysis of urea used as a cigarette additive (Baker and Bishop, 2004), oxidation of nicotine (Borduas et al., 2016a), and oxidation of formamide (Barnes et al., 2010; Borduas et al., 2015; Bunkan et al., 2015). Even the combustion of almost all sorts of common household materials including fiber glass, rubber, wood, PVC-based carpet and cables (Blomqvist et al., 2003), and polyurethane-based foam (Blomqvist et al., 2003; Jankowski et al., 2014), leads to HNCO emissions along other isocyanates (Leslie et al., 2019). HNCO emissions from traffic are originating mainly from usage of recent catalytic converters in the exhaust systems of gasoline (Brady et al., 2014) and diesel (Heeb et al., 2011) based vehicles. These converters are implemented to control the emission of primary pollutants such as hydrocarbons, carbon monoxide, particulate matter, and nitrogen oxides. However, these implementations have promoted (Suarez-Bertoa and Astorga, 2016) the formation and emissions of HNCO via surface-bound chain reactions at different stages of the flue gas exhaust, and additionally due to

emission of unreacted HNCO in the most commonly used urea-based SCR (Selective Catalytic Reduction)
system (Heeb et al., 2011). The usages of catalytic converters in modern vehicles potentially give rise to the
emission of HNCO especially in urban regions with a growing density of vehicles. A few studies also reported a
direct formation of HNCO in the diesel engines during fuel combustion without any after-treatments (Heeb et
al., 2011; Jathar et al., 2017). A tabular overview of past studies for gasoline or diesel exhaust related HNCO
emissions can be found in Wren et al. (2018) and Leslie et al. (2019). HNCO emissions via fossil fuel usage are
not limited to on-road activity. Off-road fossil fuel activities (e.g., tar sands) also contribute to significant
HNCO emissions on regional scales (Liggio et al., 2017). Finally, secondary HNCO formation in the atmosphere
is also known through the oxidation of amines and amides [e.g., (Borduas et al., 2016a; Parandaman et al.,
2017)].
The number of studies examining HNCO gas-phase chemistry are limited, and mostly focused on its
role in the chemistry in $NO_x$ mitigation strategies in combustion systems. The scarce data suggests that HNCO
destruction in the atmosphere by typical pathways such as reactions with oxidizing agents or by photolysis is
ineffective. We give a short overview here, to supplement a recent review (Leslie et al., 2019). The reaction of
HNCO with the hydroxyl radical (OH), the most important day-time oxidizing agent, has only been studied
experimentally at temperatures between 620 and 2500K (Baulch et al., 2005; Mertens et al., 1992; Tsang, 1992;
Tully et al., 1989; Wooldridge et al., 1996), where the extrapolated rate expressions lead to an estimated rate
coefficient of 5-12 $\times$ $10^{-16}$ $cm^3$ molecule$^{-1}$ s$^{-1}$ at 298 K, i.e. a HNCO-lifetime towards OH of over 25 years when
assuming a typical OH concentration of $1 \times 10^6$ molecule cm$^{-3}$. Early theoretical work by Sengupta and Nguyen
(1997) at temperatures $\geq$ 500 K showed that the mechanism proceeds predominantly by H-abstraction, forming
NCO + $H_2O$, with an energy barrier of ~6 kcal mol$^{-1}$. Wooldridge et al. (1996) determined an upper limit $\leq$ 0.1
for the fraction of $CO_2$ + $NH_2$ formation. To our knowledge, no experimental or theoretical data are available on
HNCO reactions with other dominant atmospheric oxidants, including the nitrate radical ($NO_3$), chlorine atoms
(Cl), or ozone ($O_3$). Some data is available for H- and O-atom co-reactants of importance in combustion, as well
as estimates for HCO and CN (Baulch et al., 2005; Tsang, 1992), but these are not reviewed here. There is no
direct measurement for the dry deposition of HNCO. In a global chemical transport model-based study, the
deposition velocity was considered to be similar to formic acid, yielding an HNCO-lifetime of 1 – 3 day (over
the ocean) to 1 – 2 weeks (over vegetation) (Young et al., 2012). The UV absorption for HNCO is only reported
at wavelengths < 262 nm and photolysis is mostly reported for energies at wavelengths below 240 nm by
excitation to the first singlet excited states, forming H + NCO or NH + CO (Keller-Rudek et al., 2013; Okabe,
1970; Spiglanin et al., 1987; Spiglanin and Chandler, 1987; Uno et al., 1990; Vatsa and Volpp, 2001). In the
troposphere  photolysis occurs only at UV absorption wavelength band > 290 nm due to filtering of shorter-
wavelength radiation (Hofzumahaus et al., 2002). DrozGeorget et al. (1997) have reported the photolysis of
HNCO forming NH(a$^1\Delta$) + CO(X$^1\sum^+$) at 332.4 nm, but the HNCO absorption cross-section at this wavelength
would lead to a lifetime of months (Roberts et al., 2011). Therefore, HNCO loss due to photo-dissociation
appears to be negligible in the lower atmosphere. HNCO has absorption bands in the infra-red (Sharpe et al.,
2004) but at these wavelengths the photon energy is generally too limited for photo-dissociation (Hofzumahaus
et al., 2002). The main atmospheric loss processes are considered to be the transfer to the liquid-phase followed
by hydrolysis, and deposition. This process depends on the varying atmospheric liquid water contents, relevant

temperatures, and pH of cloud droplets. Therefore, the gas-to-liquid partitioning, in the varying atmospheric properties, i.e. water content, temperature, and pH of cloud droplets become important parameters to determine the atmospheric fate of HNCO (Leslie et al., 2019). The gas-to-liquid partitioning has been described by the Henry's Law coefficient $K_H$ (ranging from 20 to 26 $\pm$ 2 M atm$^{-1}$) and related parameters by a handful of studies (Borduas et al., 2016b; Roberts et al., 2011; Roberts and Liu, 2019). Based on a recent study (Roberts and Liu, 2019), the lifetime of HNCO due to heterogeneous processes is known to be of the order of a few hours (in-cloud reactions) to weeks (aerosol deposition).

The emissions and sources of HNCO have been focused on by many past studies, but there remain large uncertainties in our understanding of HNCO removal process, especially in gas-phase chemistry. This missing information on HNCO removal processes limits global models to predict HNCO with confidence. To alleviate the dearth of direct data and therefore improve the representation of HNCO in global models, we first provide a theoretical analysis of the chemical reactions of HNCO with the dominant atmospheric oxidants: OH and $NO_3$ radicals, Cl atoms, and $O_3$ molecules, including the prediction of each rate coefficient at atmospheric conditions. In a second step, these results are included in a global numerical chemistry and climate model to assess the impact of chemical loss of HNCO in competition against hydrolysis within cloud droplets and against deposition to the Earth's surface. Additionally, the model is used to provide an estimate of the relative importance of primary and secondary HNCO sources.

## 2 Methodologies

### 2.1 Theoretical methodologies

The potential energy surfaces of the initiation reactions of all four reaction systems were characterized at the M06-2X/aug-cc-pVTZ level of theory (Dunning, 1989; Zhao and Truhlar, 2008), optimizing the geometries and rovibrational characteristics of all minima and transition states. The relative energy of the critical points was further refined at the CCSD(T) level of theory in a set of single point energy calculations using a systematic series of basis sets, aug-cc-pVxZ (x = D, T, Q) (Dunning, 1989; Purvis and Bartlett, 1982). These energies were extrapolated to the complete basis set limit (CBS) using the aug-Schwartz6(DTQ) scheme as proposed by Martin (1996). The rate coefficients were then obtained by transition state theory (Truhlar et al., 1996) in a rigid rotor, harmonic oscillator approximation, applying a scaling factor of 0.971 to the vibrational wavenumbers (Alecu et al., 2010; Bao et al., 2017). The spin-orbit splitting of the OH radicals of 27.95 cm$^{-1}$ was taken into account (Huber and Herzberg, 1979). Tunneling was incorporated using an asymmetric Eckart correction (Johnston and Heicklen, 1962).

To further complete our knowledge on some of the reactions beyond their initiation steps, the full potential energy surfaces of the HNCO + Cl and HNCO + $O_3$, were characterized at the M06-2X/aug-cc-pVTZ or B3LYP/aug-cc-pVTZ level of theory (Becke, 1993; Dunning, 1989; Lee et al., 1988), combined with CCSD(T)/aug-cc-pVTZ single point energy calculations. To our knowledge, these are the first characterizations of these surfaces. At atmospheric temperatures, most of the reaction channels are negligible, and a detailed kinetic analysis is not performed at this time.

The expected uncertainty of the rate predictions at room temperature is a factor of 4, based on an estimated uncertainty on the barrier height of at least 0.5 kcal mol$^{-1}$, and on the tunneling correction of a factor of 1.5.

Though the level of theory used is robust, there are some aspects that are not treated with the highest possible precision. For example, post-CCSD(T)/CBS calculations could refine the predicted energies, but is not expected to change our values by more than a few tenths of kcal mol$^{-1}$. The calculation of the state densities could be improved for internal rotation (especially at temperatures outside the atmospheric range), for the notoriously complex rovibronic structure of the $NO_3$ radical (Stanton, 2007, 2009; Stanton and Okumura, 2009), or by treating the transition states (micro)variationally to better characterize the energy-specific kinetic bottleneck. Another aspect is the effect of redissociation of chemically activated adducts, which decreases the effective rate of HNCO loss. Finally, tunnelling corrections for the H-abstraction reactions could benefit from higher-dimensional (curvature and corner-cutting) corrections. The tunnelling corrections are currently predicted to be smaller than a factor 15 at room temperature due to the low and broad energy barriers, except for a factor ~40 for the HNCO + $NO_3$ H-abstraction with a somewhat higher barrier. Incorporating any of the aforementioned improvements in the theoretical predictions, however, has a high to very high computational burden with strongly diminished return, as none are expected to change the rate coefficient by a factor large enough to affect the conclusions of our calculations, i.e. that the reactions are negligibly slow by many orders of magnitude compared to other HNCO loss processes (see further). This is also illustrated in Figure 2. We refer to Vereecken and Francisco (2012), Vereecken et al. (2015), and Papajakand and Truhlar (2012) for further information on theoretical methodologies in atmospheric chemistry.

## 2.2 Global modeling

The ECHAM/MESSy Atmospheric Chemistry (EMAC) model is a numerical chemistry and climate simulation system that includes sub-models describing tropospheric and middle atmosphere processes and their interaction with oceans, land and human influences (Jöckel et al., 2010). It uses the second version of the Modular Earth Submodel System (MESSy2) to link multi-institutional computer codes. The core atmospheric model is the 5th generation European Centre Hamburg general circulation model (ECHAM5) (Roeckner et al., 2006). A hierarchal diagram of EMAC is given in Jöckel et al. (2005). Additionally, Jöckel et al. (2010) provides an update on all modelling components used. For the present study we applied EMAC (ECHAM5 version 5.3.02, MESSy version 2.54.0) in the T63L90MA-resolution, i.e. with a spherical truncation of T63 (corresponding to a quadratic Gaussian grid of approximately 1.875 by 1.875 degrees in latitude and longitude) with 90 vertical hybrid pressure levels up to 0.01 hPa. By using this horizontal resolution, assessing the global impact is still feasible while at the same time being of a computationally reasonable cost. The 90 vertical layers used (focusing on the lower and middle atmosphere) represent tropospheric and stratospheric transport processes reasonable well (Jöckel et al., 2010), such that the tropospheric impact and the impact on the UTLS (upper troposphere/lower stratosphere) can be addressed. The applied model setup comprised the submodel MECCA (Module Efficiently Calculating the Chemistry of the Atmosphere) to calculate atmospheric chemistry using parts of the Mainz Organic Mechanism (MOM) (Sander et al., 2011). Within MOM, aromatics and terpenes were excluded to reduce the computational demand of all simulations performed; this chemistry has no relevant impact on HNCO. The mechanism was extended to include the proposed changes of this study, formamide as an additional chemical source of HNCO (Bunkan et al., 2016), and chemical mechanisms for nitromethane (Calvert, 2008; Taylor et al., 1980), methylamine, dimethylamine and trimethylamine (Nielsen et al., 2012). The reaction rates used for the latter three are average values of the measured values reported in Nielsen et al.

(2012). The product yields reported in the same source are simplified to suit a global model application. The submodel SCAV (SCAVenging submodel) was used to simulate the physical and chemical removal of trace gases and aerosol particles by clouds and precipitation (Tost et al., 2006). The aqueous phase mechanism was extended to include the HNCO and formamide mechanism proposed by Borduas et al. (2016b), Barnes et al. (2010), and Behar (1974). These lead to the formation to ammonia in the aqueous-phase, which was before limited to the acid-base equilibrium in cloud droplets. The representation of cyanide was improved based on Buechler et al. (1976). Table 1 and 2 in the supplementary material summaries all additional changes to the chemical mechanism in gas and aqueous phase, respectively. The submodel DDEP (Dry DEPosition) is used to simulate the dry deposition of HNCO, using the default scheme with non-stomatal uptake effectively not being considered by mean of a large and constant resistance (Kerkweg et al., 2006a). The effective Henry's law coefficient (H*) is used, as proposed by Borduas et al. (2016b), modified to a pH of 7. Differently from Young et al. (2012), the same H* over the ocean is used. This approximation is reasonable since the levels of HNCO in the marine boundary layer are expected to be minor. In a global context, the major sources of HNCO and formamide are biomass burning emissions. From literature two emission factors are available, which differ substantially: 0.53 g kg$^{-1}$ (Koss et al., 2018) versus 0.2 g kg$^{-1}$ (Kumar et al., 2018). Thus two simulations are performed, to quantify the uncertainty in those emission factors. The MESSy submodel BIOBURN is used to calculate biomass burning fluxes based on the selected emission factor and Global Fire Assimilation System (GFAS) data. GFAS data are calculated based on fire radiative power observations from Moderate Resolution Imaging Spectroradiometer (MODIS) satellite instruments, which are used to calculate the dry matter combustion rates (Kaiser et al., 2012). The biomass burning emission fluxes are then obtained by combining these dry matter combustion rates with the defined biomass burning emission factors per unit of dry matter burned. The MESSy submodel OFFEMIS (OFFline EMISsions) then calculates the resulting concentration changes for each tracer due to the biomass burning emissions (Kerkweg et al., 2006b). Anthropogenic HNCO emission from diesel cars are scaled to ammonia EDGAR (Crippa et al., 2016) road emission by 15% (Heeb et al., 2011). Other known sources of HNCO (e.g. cigarette smoke) were not taken into account due to the resolution of the spatial grid used. The model was run for two years (2010-2011) in which the first year was used as spin up and 2011 for analysis. In 2010, the biomass burning emissions were particular high (Kaiser et al., 2012), providing higher background HNCO concentrations during spin up, improving the representation of HNCO which allows for a more representative comparison in 2011.

**3 Loss processes by chemical oxidants**

**3.1 HNCO + OH**

The reaction of HNCO with OH can proceed by 4 distinct pathways: H-abstraction, or OH addition on the carbon, nitrogen, or oxygen atom of HNCO; a potential energy surface is shown in Figure 1. Formation of the HN=C$^•$OOH and HN(OH)C$^•$=O adducts through OH-addition on the oxygen or nitrogen atom is highly endothermic by 20 kcal mol$^{-1}$ or more, and is not competitive at any temperature. The two remaining pathways are exothermic, with HN$^•$C(=O)OH being the most stable nascent product, 19.8 kcal mol$^{-1}$ below the reactants, followed by H$_2$O + $^•$N=C=O, at 7.5 kcal mol$^{-1}$ exoergicity. Despite the higher energy of the products, we predict this latter reaction to have a lower barrier, 6.0 kcal mol$^{-1}$, compared to the addition process, 8.7 kcal mol$^{-1}$, in

agreement with the theoretical predictions of Sengupta and Nguyen (1997). Furthermore, the H-abstraction
process allows for faster tunneling, making this process the fastest reaction channel, while addition contributes
less than 0.5% of product formation at temperatures below 400K. From this data, we derive the following rate
coefficient expressions (see also Figure 2):

5        $k_{OH}(298K) = 7.03 \times 10^{-16}$ cm$^3$ molecule$^{-1}$ s$^{-1}$

6        $k_{OH}(200\text{-}450K) = 3.27 \times 10^{-34}$ T$^{7.01}$ exp(685K/T) cm$^3$ molecule$^{-1}$ s$^{-1}$

7        $k_{OH}(300\text{-}3000K) = 1.79 \times 10^{-23}$ T$^{3.48}$ exp(-733K/T) cm$^3$ molecule$^{-1}$ s$^{-1}$

Our predictions are in very good agreement between 624-875K, when compared with experimental data from
Tully et al. (1989), which served as the basis for the recommendation of Tsang (1992); our predictions
reproduce the rate coefficients within a factor 1.7, comparable to the experimental uncertainty of a factor 1.5
(see Figure 2). Likewise, our predictions agree within a factor 1.7 with the experimental determination of
Wooldridge et al. (1996), over the entire 620-1860 K temperature range. Our predictions overshoot the upper
limit estimated by Mertens et al. (1992) by a factor of up to 4 at the upper end of the temperature range (2120 to
2500 K). At these elevated temperatures, it is expected that our theoretical kinetic calculations are less accurate
since anharmonicity, internal rotation, and possibly pressure effects are not fully accounted for. At this time, we
choose not to invest the computational cost to improve the predictions at these temperatures. The predicted rate
at room temperature is within a factor of 2 of the extrapolation of the recommended expression derived by
Tsang (1992), $k(298 \text{ K}) \approx 1.24 \times 10^{-15}$ cm$^3$ molecule$^{-1}$ s$^{-1}$, and very close to the extrapolation of the expression by
Wooldridge et al. (1996), $7.2 \times 10^{-16}$ cm$^3$ molecule$^{-1}$ s$^{-1}$. The good agreement of our rate coefficient with the
experimental data extrapolated to room temperature is mainly due to the curvature predicted in the temperature-
dependence (see Figure 2), as our calculations have a slightly steeper temperature dependence than the
experiments in the high-temperature range. Though negligible at low temperature, we find that OH addition on
the C-atom of HNCO accounts for 7 to 8 % of the reaction rate between 2000 and 3000 K, with other non-H-
abstraction channels remaining negligible (<0.1%). The addition channel is the likely origin of $CO_2$ + $NH_2$
products (Sengupta and Nguyen, 1997), for which Wooldridge et al. (1996) experimentally determined an upper
limit ≤ 0.1 over the temperature range 1250-1860 K, corroborating our predictions to its low contribution.
Typical concentrations of the OH radical during daytime are measured at ~$10^6$ molecule cm$^{-3}$ (Stone et al.,
2012), leading to a pseudo-first order rate coefficient for HNCO loss by OH radicals of $k(298K) = 7 \times 10^{-10}$ s$^{-1}$,
i.e. suggesting an atmospheric chemical lifetime of decades to several centuries, depending on local temperature
and OH concentration, negligible compared to other loss processes like scavenging. Even in extremely dry
conditions, where aqueous uptake is slow, heterogeneous loss processes will dominate, or alternatively
atmospheric mixing processes will transport HNCO to more humid environments where it will hydrolyze.
**3.2 HNCO + Cl**
From the potential energy surface (PES) shown in Figure 1, we see that the reaction between HNCO and Cl
atom can occur by abstraction of the H atom from HNCO, or by addition of the Cl atom on the C-, N- or O-
atoms. Contrary to the OH-reaction, all entrance reactions are endothermic, with formation of the HN$^\bullet$C(Cl)=O
alkoxy radical nearly energy-neutral (see Figure 1). Formation of this latter product, proceeding by the addition
of a Cl atom to the carbon atom of HNCO, also has the lowest energy barrier, 7.3 kcal mol$^{-1}$ above the reactants.

The hydrogen abstraction, forming HCl and $^\bullet$NCO, requires passing a higher barrier of 11.2 kcal mol$^{-1}$, whereas addition on the N- and O-atoms have very high barriers exceeding 34 kcal mol$^{-1}$. The product energy difference between addition and H-abstraction is much smaller compared to the HNCO + OH reaction. Despite this reduced reaction energy, the addition barrier remains 4 kcal mol$^{-1}$ below the H-abstraction barrier, making the HNCO + Cl reaction the only reaction studied here where H-abstraction is not dominant. For the HNCO + Cl reaction, we then obtain the following rate coefficients (see also Figure 3):

$$k_{Cl}(298K) = 3.19 \times 10^{-17} \text{ cm}^3 \text{ molecule}^{-1} \text{ s}^{-1}$$

$$k_{Cl}(200\text{-}450K) = 1.11 \times 10^{-17} \text{ T}^{1.97} \exp(-3031K/T) \text{ cm}^3 \text{ molecule}^{-1} \text{ s}^{-1}$$

We find that the overall rate coefficient of the HNCO + Cl reaction is almost one order of magnitude below that for the OH radical. The HN$^\bullet$C(Cl)=O radical formed, however, has a weak C–Cl bond requiring only 5.4 kcal mol$^{-1}$ to redissociate. The rate coefficient of $8 \times 10^8$ s$^{-1}$ for dissociation at room temperature ($k(T) = 8.3 \times 10^{12}$ exp(-2760/T) s$^{-1}$), is over an order of magnitude faster than O$_2$ addition under atmospheric conditions, assuming the latter is equally fast as for H$_2$C$^\bullet$CH=O vinoxy radicals, i.e. k(298K, 0.2 atm O$_2$) $\leq 10^7$ s$^{-1}$ (IUPAC Subcommittee on Atmospheric Chemical Kinetic Data Evaluation, 2017). This makes redissociation to the reactants the most likely fate of the HN$^\bullet$C(Cl)=O adduct. Addition is thus an ineffective channel for HNCO removal, and the effective reaction with Cl atoms is dominated by the H-abstraction reaction, forming HCl + $^\bullet$NCO, with the following rate coefficient (see also Figure 3):

$$k_{Cl,eff}(298K) = 2.23 \times 10^{-19} \text{ cm}^3 \text{ molecule}^{-1} \text{ s}^{-1}$$

$$k_{Cl,eff}(200\text{-}450K) = 1.01 \times 10^{-24} \text{ T}^{4.40} \exp(-3799 \text{ K}/T) \text{ cm}^3 \text{ molecule}^{-1} \text{ s}^{-1}$$

Globally, Cl atoms have a lower concentration, about $5 \times 10^3$ atom cm$^{-3}$, compared to OH radicals (Finlayson-Pitts and Pitts, 1999). Under such conditions, lifetimes estimated for HNCO towards Cl atoms are about $3 \times 10^7$ years, which is much longer than toward the OH radial. Therefore, HNCO loss by Cl radicals is negligible.

The supporting information provides information on the extended potential energy surface of the HNCO + Cl reaction, with information on 9 intermediates, 19 transition states, and 16 products.

## 3.3 HNCO + NO$_3$

The reaction of NO$_3$ with HNCO shows the same four radical mechanisms found for OH and Cl, i.e. H-abstraction and addition on the 3 heavy atoms. As for Cl-atoms, none of the reactions are exothermic, and the energy difference between the two most stable products is reduced to 3 kcal mol$^{-1}$, indicating that NO$_3$ addition is even less favorable than Cl addition. Formation of HNO$_3$ + $^\bullet$NCO is more favorable than HCl + NCO formation, by about 2 kcal mol$^{-1}$. The barrier for H-abstraction, however, is larger compared to abstraction by both OH and Cl, and exceeds 12 kcal mol$^{-1}$. The most favorable addition process, forming HN$^\bullet$C(=O)NO$_3$ has a barrier of 15.1 kcal mol$^{-1}$, but contributes less than 0.01% to the reaction rate at room temperature. The overall reaction thus proceeds near-exclusively by H-abstraction forming HNO$_3$ + $^\bullet$NCO, for which we derived the following rate coefficients (see also Figure 3):

$$k_{NO3}(298K) = 1.11 \times 10^{-21} \text{ cm}^3 \text{ molecule}^{-1} \text{ s}^{-1}$$

$$k_{NO3}(200\text{-}450K) = 8.87 \times 10^{-42} \text{ T}^{9.06} \exp(-1585K/T) \text{ cm}^3 \text{ molecule}^{-1} \text{ s}^{-1}$$

While this rate coefficient is almost 5 orders of magnitude below that of the OH radical, the nitrate radical is known to be present in higher concentrations during night time, reaching concentrations as high as $10^9$ molecule

cm$^{-3}$ (Finlayson-Pitts and Pitts, 1999). The effective rate of the $NO_3$ reaction at night time is similar to the
reaction with OH at day time. The $NO_3$ radical is thus likewise considered to be ineffective for atmospheric
removal of HNCO, compared to heterogeneous loss processes.
**3.4 HNCO + O$_3$**
The chemistry of ozone with organic compounds is drastically different from radicals, where $O_3$ typically reacts
by cycloaddition on double bonds in unsaturated compounds. For HNCO, cycloaddition pathways have been
characterized for both double bonds (HN=C=O). Only cycloaddition on the N=C bond leads to an exothermic
reaction, with the oxo-ozonide product being 12 kcal mol$^{-1}$ more stable than the reactants (see Figure 1). In
addition to the traditional cycloaddition channels, three further channels were found, corresponding to H-
abstraction, forming $HO_3$ + NCO, oxygen transfer to the N-atom, forming ON(H)CO + $^1O_2$, and addition on the
C- and N-atom, forming HN(OO)C(O)O. The $HO_3$ product radical is known to be only weakly bonded by 2.94
kcal mol$^{-1}$, falling apart to OH + $O_2$ (Bartlett et al., 2019; Le Picard et al., 2010; Varandas, 2014).
The cyclo-addition channels on the hetero-double bonds have high energy barriers, exceeding 30 kcal mol$^{-1}$,
significantly larger than typical barriers for C=C bonds with aliphatic substitutions. Surprisingly, this allows H-
abstraction to become competitive to cycloaddition, with a comparable barrier of 32 kcal mol$^{-1}$. For the overall
reaction, we obtain the following rate coefficients (see also Figure 3):

17            $k_{O_3}(298K) = 2.95 \times 10^{-37}$ cm$^3$ molecule$^{-1}$ s$^{-1}$

18            $k_{O_3}(200\text{-}450K) = 3.72 \times 10^{-23}$ T$^{2.96}$ exp(-14707K/T) cm$^3$ molecule$^{-1}$ s$^{-1}$

At room temperature, H-abstraction contributes 80% to the total reaction, and cycloaddition on the N=C bond
the remaining 20%. All other channels are negligible. The rate coefficient is exceedingly low, ~$10^{-37}$ cm$^3$
molecule$^{-1}$ s$^{-1}$, such that even in areas with very high ozone concentrations of 100 ppbv the loss by ozonolysis is
expected to be negligible.
The supporting information provides information on the extended potential energy surface of the HNCO + $O_3$
reaction, with information on 10 intermediates, 30 transition states, and 15 products. The lowest-energy
unimolecular product channel leads to formation of $CO_2$ + HNOO by breaking of the cyclic primary ozonide
(see Figure 1) following the traditional Criegee mechanism (Criegee, 1975).
**4 H-abstraction reactions by NCO radicals**
The radical reactions characterized above proceed by H-abstraction, forming the NCO radical with an $H_2O$,
$HNO_3$, or HCl co-product. Likewise, the ozonolysis reaction proceeds for a large part by H-abstraction, forming
NCO with a $HO_3$ coproduct that readily dissociates to OH + $O_2$. Though NCO radical formation through these
reactions is found to be negligibly slow in atmospheric conditions, this radical remains of interest in other
environments. Examples include combustion chemistry, where it can be formed directly from nitrogen-
containing fuels, and where it is a critical radical intermediate in e.g. the RAPRENOx nitrogen-oxide mitigation
strategy which employs HNCO introduced in the combustion mixture through (HOCN)$_3$ (cyanuric acid)
injection (Fenimore, 1971; Gardiner, 2000). The NCO radical has also been observed in space (Marcelino et al.,
2018). There is extensive experimental and theoretical information of the reactions of NCO radicals, tabulated
e.g. in Tsang (1992), Baulch et al. (2005) and other works. To our knowledge, the rate coefficients of the
reactions of NCO radicals with $H_2O$, $HNO_3$, and HCl have not been determined before, though Tsang (1992)
has estimated a rate coefficient k(NCO + $H_2O$) = $3.9 \times 10^{-19}$ $T^{2.1}$ exp(-3046K/T) $cm^3$ molecule$^{-1}$ s$^{-1}$ based on the
equilibrium constant and rate coefficient of the HNCO + OH reaction. Since the H–N bond in HNCO is quite
strong, with a bond energy of ~110 kcal/mol (Ruscic, 2014; Ruscic and Bross, 2019), it is expected that NCO
can readily abstract a hydrogen atom from most hydrogen-bearing species to produce HNCO, and that H-
abstraction is the main reaction channel. Hence, despite that our potential energy surfaces do not include an
exhaustive search of all possible reaction channels in the NCO radical chemistry, we expect that the single-
channel H-abstraction rate predictions for NCO from $H_2O$, $HNO_3$ and HCl is sufficiently dominant that these
rates are fair estimates of the total rate coefficients including all possible channels for each of these reactions.
The energy barriers for the NCO radical reactions with $H_2O$, $HNO_3$ and HCl, being 14, 7, and 4 kcal mol$^{-1}$
respectively (see Figure 1), follow the bond strength trend in these reactants, with $D_0$(H–OH) = 118 kcal mol$^{-1}$,
$D_0$(H–$NO_3$) = 104 kcal mol$^{-1}$, and $D_0$(H–Cl) = 103 kcal mol$^{-1}$ (Luo, 2007; Ruscic et al., 2002). Figure 1 also
shows that the NCO + $H_2O$ reaction is endothermic by 8 kcal mol$^{-1}$, while the $HNO_3$ and HCl paths are
exothermic by -5 and -7 kcal mol$^{-1}$, respectively. The predicted rate coefficients are then:
$\qquad$ $k_{NCO+H2O}$(300K) = $1.36 \times 10^{-21}$ $cm^3$ molecule$^{-1}$ s$^{-1}$
$\qquad$ $k_{NCO+HNO3}$(300K) = $3.37 \times 10^{-17}$ $cm^3$ molecule$^{-1}$ s$^{-1}$
$\qquad$ $k_{NCO+HCl}$(300K) = $1.39 \times 10^{-14}$ $cm^3$ molecule$^{-1}$ s$^{-1}$
$\qquad$ $k_{NCO+H2O}$(300-3000K) = $4.59 \times 10^{-24}$ $T^{3.63}$ exp(-4530K/T) $cm^3$ molecule$^{-1}$ s$^{-1}$
$\qquad$ $k_{NCO+HNO3}$(300-3000K) = $7.18 \times 10^{-26}$ $T^{4.21}$ exp(-1273K/T) $cm^3$ molecule$^{-1}$ s$^{-1}$
$\qquad$ $k_{NCO+HCl}$(300-3000K) = $3.73 \times 10^{-20}$ $T^{2.63}$ exp(-662K/T) $cm^3$ molecule$^{-1}$ s$^{-1}$
The indirect estimate of Tsang (1992) compares well to our prediction for NCO + $H_2O$, reproducing our values
within a factor 15 at 1000K and factor 3 at 2000K, i.e. within the stated uncertainties. An analysis of the impact
of the NCO reactions in combustion or non-terrestrial environments is well outside the scope of this paper, and
reactions with other co-reactants not discussed in this paper are likely to be of higher importance, e.g. H-
abstraction from organic compounds, or recombination with other radicals. In atmospheric conditions, the fate
of the NCO radical is likely recombination with an $O_2$ molecule, leaving $H_2O$, $HNO_3$, and HCl as negligible co-
reactants. Hence, the NCO radical will not affect the atmospheric fate of any of these compounds to any extent.
Subsequent chemistry of the $^\bullet$OONCO radical is assumed to be conversion to an $^\bullet$ONCO alkoxy radical through
reactions with NO, $HO_2$ or $RO_2$, followed by dissociation to NO + CO.

## 5 Global impact

Global atmospheric simulations allow us to gain insights on the significance of the chemical loss processes of
HNCO and its distribution. Table 1 shows the corresponding HNCO budget for both performed simulations. The
full kinetic model including our theoretically predicted gas-phase chemical reactions of HNCO is detailed in
Table 1 and 2 of the supplementary material. Figure 4 shows the mean seasonal surface mixing ratio of HNCO
using the biomass burning emission factors by Koss et al. (2018). It can be observed that high levels persist in
each season. In general, high HNCO levels occur in regions associated with frequent biomass burning activities.
Regions with no biomass burning activities have low HNCO concentrations, mainly caused by free tropospheric
entrainment from regions with higher concentrations. The global vertical profile of HNCO is well illustrated by

that for January as given in Figure 5, showing that the free troposphere contains about 81% of the total HNCO mass. The gas-phase production via formamide differs greatly depending on the biomass burning emissions used. In the case of Kumar et al. (2018), significantly more formamide is emitted, leading to a higher production of HNCO in the gas phase. The hydrolysis of HNCO produces ~120 Tg/yr of ammonia, thus contributing little to the global ammonia budget. Our estimate is a factor 5-6 lower than the upper limit estimated by Leslie et al. (2019).

The model predictions for OH radical concentrations range from $1.15 \times 10^0$ to $1.56 \times 10^7$ molecule cm$^{-3}$, with a weighted atmospheric global average of $1.14 \times 10^6$ molecule cm$^{-3}$; in the air parcel where the highest OH concentration is found this leads to an HNCO lifetime towards OH of more than 500 years when accounting for the temperature-dependent rate coefficient (~276K). In the planetary boundary layer, the highest OH concentration predicted is $7.6 \times 10^6$ molecule cm$^{-3}$ at a temperature of 297.8K, leading to an HNCO lifetime to OH of ~6 years in that air parcel. The calculated average OH concentration of $1.20 \times 10^6$ molecule cm$^{-3}$ in the boundary layer leads to lifetimes towards OH of about 40 years near the surface. For O$_3$, Cl, and NO$_3$, with maximum oxidant concentrations of $1.0 \times 10^{13}$, $7.8 \times 10^5$, $1.5 \times 10^9$ molecule cm$^{-3}$, and atmospheric average concentrations of $1.0 \times 10^{12}$, $2.0 \times 10^3$, $1.1 \times 10^7$ molecule cm$^{-3}$, respectively, even longer temperature-dependent lifetimes are found, exceeding 5000 years even in the airparcels with the most favourable co-reactant concentration and temperature. The relative contributions of the different co-reactants varies locally and temporally, and shorter lifetimes might occur locally when co-reactant concentration and temperature are at their most favourable, but it is clear that gas phase chemical losses of HNCO are small. Only the reaction of HNCO with OH leads to some destruction of HNCO, while the other chemical sinks (O$_3$, NO$_3$ and Cl) are negligible. When compared to the major loss processes, however, all these loss processes are on a global scale negligible (see Table 1). Young et al. (2012) have a somewhat higher chemical loss via OH compared to our result, which is due to the higher rate constant used. Figure 2 shows the rate coefficient that would be required to allow the gas phase loss of HNCO by reaction with OH radicals to contribute 10% of the total atmospheric sink, which is well outside the expected uncertainty of the theoretical kinetic rate predictions. It can therefore be robustly concluded that the gas-phase chemical sinks predicted and assessed in this study (OH, Cl, NO$_3$, O$_3$) are insignificant when compared to heterogeneous loss processes, confirming earlier assumptions. This is independent of the high uncertainty in the available biomass burning emission factors or missing road emission datasets.

As seen in Table 1 the major sinks are dry deposition and scavenging (heterogeneous losses), where the former contributes between 2519.61 and 2891.85 Gg/year, and the latter from 274.60 to 377.19 Gg/year, when using the emission factors by Koss et al. (2018) and Kumar et al. (2018), respectively. The results in this study are in a similar range as the modelling study by Young et al. (2012). These authors had lower total HNCO emissions and did not include formamide as a secondary source of HNCO. The lower total HNCO emissions could be explained by a different year simulated in that study and different biomass burning emission model approaches used. Young et al. (2012) also scaled their HNCO emissions to the HCN emissions by a factor of 0.3, whereas in this study actual measured emission factors are used. In our study, formamide contributes between 17.16% and 70.46% of the total HNCO emissions when using the biomass burning emission factors by Koss et al. (2018) and Kumar et al. (2018) respectively. Young et al. (2012) find a higher HNCO lifetime due to generally lower total heterogeneous loss terms (dry and wet deposition). The total dry deposition varies slightly depending on

the biomass burning emission factor used (see Table 1). In both scenarios, most HNCO is deposited over the ocean. For biomass burning emission factors from Koss et al. (2018) this contribution, 53.3%, is significantly lower when compared to the simulation using emission factors from Kumar et al. (2018), where about 62.5% of the total HNCO deposition is deposited over the ocean. The larger fraction of computed HNCO deposition over the ocean is a consequence of the much larger secondary HNCO production from formamide far from its source regions (continents). Young et al. (2012) found that the importance of both heterogeneous loss processes depends on the clouds pH. In the SCAV submodel, as used in this work, cloud droplet pH is calculated online and includes an explicit hydrolysis scheme for HNCO, whereas Young et al. (2012) used a simplified approach. The relative importance of dry deposition is higher in the simulation in which Young et al. (2012) calculated pH online, when compared to the findings in this study.

The atmospheric lifetime of HNCO is dominated by its heterogeneous loss processes, leading to an atmospheric lifetime of multiple weeks when accounting for all HNCO losses (chemical and heterogeneous), as opposed to a gas-phase lifetime in the free troposphere of about 50 years when calculated solely based on the chemical losses towards the four chemical oxidants described in this study. This long gas-phase lifetime and the fact that mainly surface sources are relevant indicate that atmospheric HNCO distribution is significantly affected by transport processes. Our simulations even show that HNCO is transported from the surface into the UTLS and that about 10% of the total atmospheric HNCO mass is located in the stratosphere (see Figure 5), with modelled concentrations of HNCO in the lower stratosphere of typically tens of pptv but reaching up to hundred pptv in tropical regions. In the chemical model, photolysis in the stratosphere was not taken into account. Thus, OH is the only significant stratospheric sink included, resulting in a stratospheric lifetime of more than 330 years. During the monsoon period, the total stratospheric HNCO mass increases from 15.04 Gg before, to 19.75 Gg at the end of monsoon season. Pumphrey et al. (2018) demonstrated that in 2015 and 2016, elevated levels of stratospheric hydrogen cyanide (HCN) can be linked to biomass burning emissions from Indonesian fires. Figure 5 shows the vertical profiles of HCN and HNCO over South East Asia well before (January) and after (November) the Indian monsoon. It becomes evident that, similar to HNCO in our simulations, tropospheric and stratospheric concentrations of HCN increase during the Indian monsoon period. In the performed simulations, the ratio between stratospheric HCN and HNCO is very similar throughout the year, indicating that HCN and HNCO are similarly affected by transport processes within this period. The combination of strong biomass burning events and strong vertical transport during the monsoon period leads to high HNCO concentrations in the UTLS, indicating that pollutants from biomass burning events could potentially influence stratospheric chemistry.

Figure 6 shows the number of days exceeding a daily mean HNCO concentration of 1 ppbv. Mainly regions impacted by biomass burning events have frequent concentration above this threshold. When using 10 ppbv as limit for toxic concentrations of HNCO, as proposed by the Swedish work environment authority (SWEA, 2011), only a few days can be observed in which this limit is exceeded. The maximum number of days exceeding 10 ppbv is 10 days over Africa, compared to 120 days above 1 ppbv. It is important to take into account that this analysis is limited by the computational output available in this study, which has only daily averages. Therefore, it is expected that areas which frequently exceed daily averages of 1 ppbv are potentially areas in which peak HNCO can be observed above 10 ppbv throughout the day.

No correlation exists between the number of days exceeding 1 or 10 ppbv and road traffic emissions. This becomes evident since typical areas of high road traffic activities (i.e. USA and Europe) do not exceed daily averages of 1 ppbv (seeFigure 6). Road traffic activities occur on a smaller spatial scale than biomass burning events. The EMAC model used is not capable to represent, for example, inner city road traffic activities, due to the spatial resolution of the model used (1.875 by 1.875 degrees in latitude and longitude). Therefore, we are not capable to draw any conclusion if 10 ppbv is exceeded regionally in densely populated areas, impacted by high traffic emissions.

## 6 Conclusions

The isocyanic acid molecule, HNCO, is found to be chemically unreactive towards the dominant atmospheric gas phase oxidants, i.e. OH and $NO_3$ radicals, Cl atoms, and $O_3$ molecules. The reactions all remove HNCO predominantly by H-abstraction, and have low rates of reactions with $k(298) \leq 7 \times 10^{-16}$ $cm^3$ $molecule^{-1}$ $s^{-1}$, leading to chemical gas phase lifetimes of decades to centuries. Yearly loss of HNCO towards these reactants is only ~5 Gg/y out of ~3000 Gg/y total losses. Removal of HNCO by clouds and precipitation ("scavenging"), with hydrolysis to ammonia, is also implemented in the global model, and was found to contribute significantly more, ~300 Gg/y, than the gas phase loss processes. Still, these combined processes are overwhelmed by the loss of HNCO by dry deposition, removing ~2700 Gg/y. These conclusions are robust against modifications of the emission scenarios, where two distinct sets of emission factors were used, incorporating HNCO formation from biomass burning, as well as anthropogenic sources such as formamide oxidation and road traffic. The inefficiency of gas-phase chemical loss processes confirms earlier assumptions; inclusion of the gas-phase chemical loss processes in kinetic models appears superfluous except in specific experimental conditions with very high co-reactant concentrations. The long gas-phase chemical lifetime (multiple decades to centuries) allows HNCO to be transported efficiently into the UTLS demonstrating that surface emissions may impact the upper troposphere. Further research is necessary to identify the importance of strong biomass burning events coupled to strong vertical transport processes (i.e. monsoon systems) on the chemical composition of the UTLS. On a global scale, the daily average concentrations of HNCO rarely exceed 10 ppbv, the threshold assumed here for toxicity; the exceedances are mainly located in regions with strong biomass burning emissions. Average daily concentrations of the order of 1 ppbv are encountered more frequently, with about 1/3th of the year exceeding this limit. This suggests that local concentrations might peak to much higher values, e.g. in urban environments where road traffic emissions are highest, or in the downwind plume of biomass burning events, and could impact regional air quality. Such regional effects were not studied in the current work, as the resolution of the global model used here is not sufficiently fine-grained.

Though not important for the atmosphere, we briefly examined the reactions of the NCO radical formed in the chemical reactions studied. The rate coefficients of the H-abstraction reactions with $H_2O$, $HNO_3$ and HCl suggest that these reactions might contribute in high-temperature environments, such as combustion processes.

## Supplement

The supplement related to this article is available online, and contains extended information on the chemical model, and the quantum chemical characterizations (geometric, energetic and entropic data)

## Author contributions

The quantum chemical calculations where performed by H.M.T. Nguyen, G.H.T. Vu, and T.V. Pham, while L. Vereecken performed the theoretical kinetic calculations. U. Javed, S. Rosanka and D. Taraborrelli collected the literature data on HNCO sources and sinks, and implemented these in the kinetic model; the model runs were performed by S. Rosanka and D. Taraborrelli. All authors contributed significantly to the writing of the manuscript.

## Competing interests

The authors declare that they have no conflict of interest.

## Acknowledgments

HMTN, GHTV and TVP thank the National Foundation for Science and Technology Development (Nafosted), Vietnam for sponsoring this work under project number 104.06-2015.85. SR and DT gratefully acknowledge the Earth System Modelling Project (ESM) for funding this work by providing computing time on the ESM partition of the supercomputer JUWELS at the Jülich Supercomputing Centre (Forschungszentrum Jülich, 2019).

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

**Table 1: Yearly global HNCO budget in 2011 for both biomass burning emission datasets by Kumar et al. (2018) and Koss et al. (2018). Additionally, the HNCO budget from Young et al. (2012) is given for comparison.**

| | Simulations in this study based on emission factors from: | | Comparable literature: |
|---|---|---|---|
| | **Koss et al., 2018** | **Kumar et al., 2018** | **Young et al., 2012** |
| *Emissions [Gg/year]* | | | |
| Biomass burning (HNCO) | 2158.94 | 814.69 | 661.00 |
| Anthropogenic (HNCO) | 177.14 | 177.14 | 828.00 |
| *Gas phase production [Gg/year]* | | | |
| $NH_2CHO + OH$ | 482.52[a] | 2365.53[b] | - |
| *Gas phase loss [Gg/year]* | | | |
| $HNCO + OH$ | 3.98 | 5.41 | ~ 5.96 |
| $HNCO + O_3$ | $1.88 \times 10^{-16}$ | $2.37 \times 10^{-16}$ | - |
| $HNCO + NO_3$ | $1.15 \times 10^{-4}$ | $1.43 \times 10^{-4}$ | - |
| $HNCO + Cl$ | $9.99 \times 10^{-8}$ | $1.37 \times 10^{-7}$ | - |
| *Heterogeneous losses [Gg/year]* | | | |
| Dry deposition | 2519.61 | 2891.85 | ~ 1421.99 |
| Over land | 1174.92 | 1086.13 | - |
| Over ocean | 1344.69 | 1805.72 | - |
| Scavenging | 274.60 | 377.19 | - |
| Wet deposition | 0.13 | 0.16 | ~ 67.01 |
| Yearly mean burden [Gg] | 201.15 | 271.94 | ~ 150.00 |
| Atmospheric lifetime [days] | 26.24 | 30.31 | 36.62 |

5    [a] Of which 50.59 Gg/year $NH_2CHO$ biomass burning emissions (Koss et al., 2018)

6    [b] Of which 2335.01 Gg/year $NH_2CHO$ biomass burning emissions (Kumar et al., 2018)

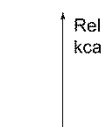

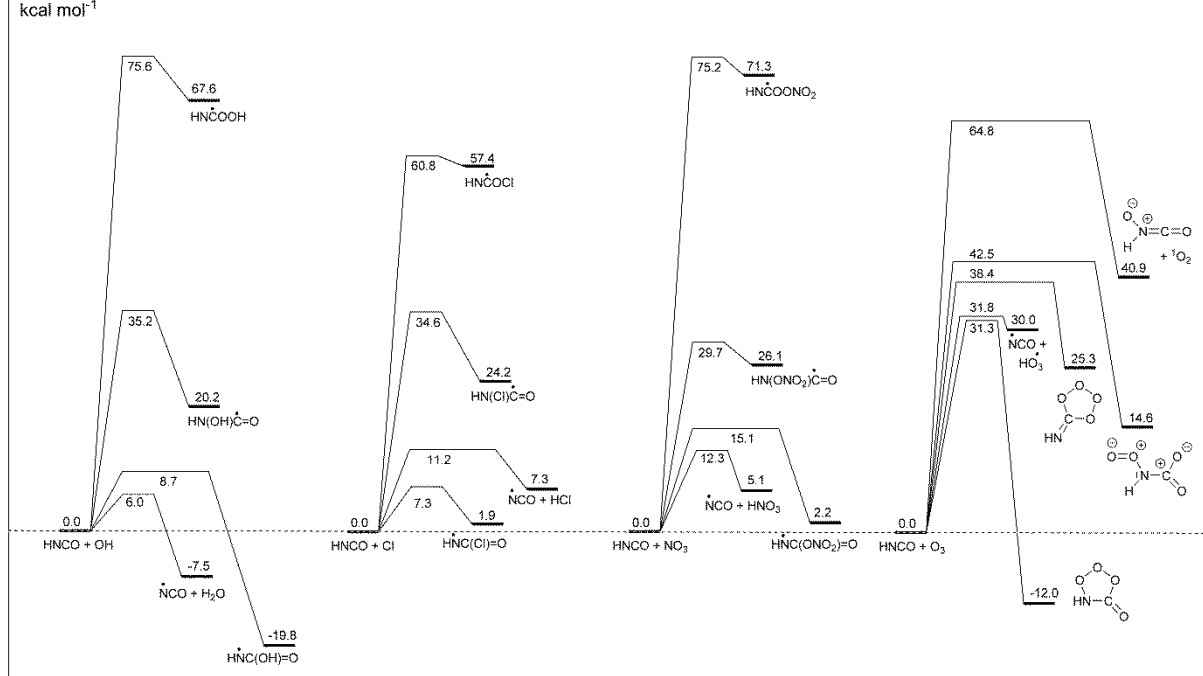

**Figure 1: Potential energy surfaces for the initiation reactions of HNCO with OH radicals, Cl atoms, NO₃ radicals,**
**and ozone, showing CCSD(T)/CBS(DTQ) energies (kcal mol⁻¹) based on M06-2X/aug-cc-pVTZ geometries. The pre-**
**reactive complexes are omitted as they do not influence the kinetics,; similarly, the subsequent reactions of the**
**products are not shown. The supporting information has additional energetic and rovibrational data, more complete**
**potential energy surfaces for some of the reactions, as well as three-dimensional representations of the molecular**
**structure with bond lengths and angles.**

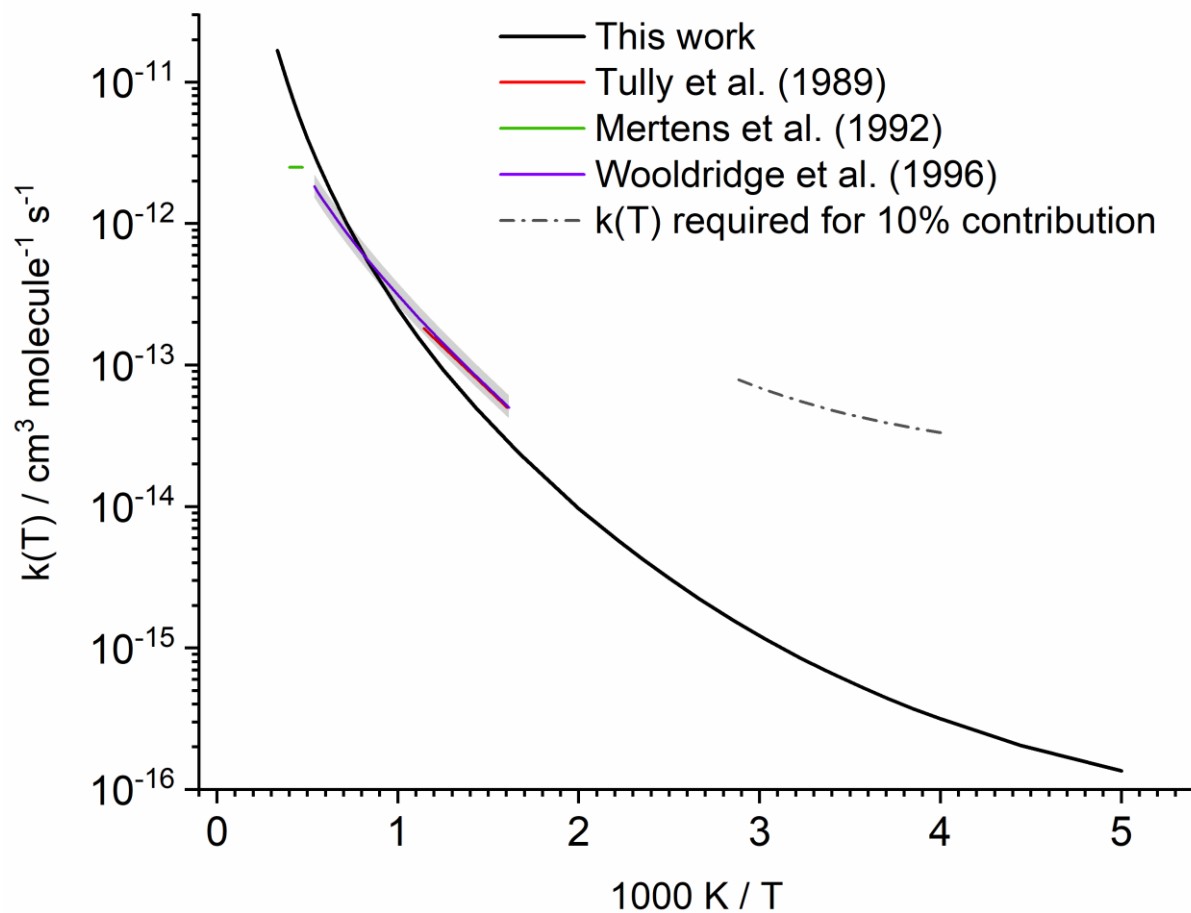

**Figure 2: Predicted rate coefficient k(T) for the reaction of HNCO + OH, compared against experimental data. The**
**shaded area indicates the experimental uncertainty reported by Wooldridge et al. (1996). The dashed line estimates**
**the 298 K rate coefficient that would be needed to remove 10% of the atmospheric HNCO by reaction with OH (see**
**text).**

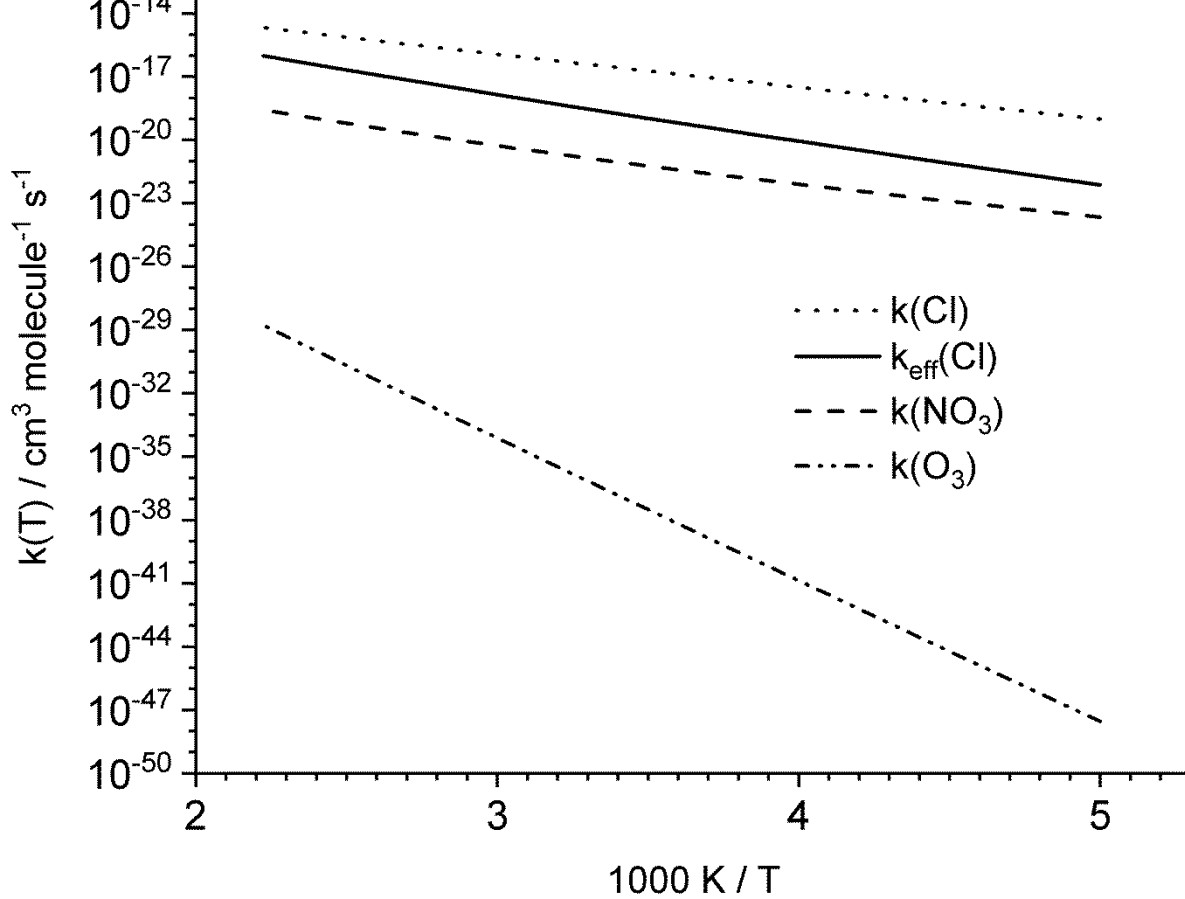

**Figure 3: Total rate coefficient predictions for the reaction of HNCO with $NO_3$, Cl and $O_3$. The addition of Cl atoms**
**on HNCO leads to the formation of a very short-lived adduct, which rapidly redissociates to the reactants; the**
**effective rate coefficient for HNCO loss by Cl atoms, $k_{eff}$(Cl), is thus equal to the H-abstraction rate forming HCl +**
**NCO (see text).**

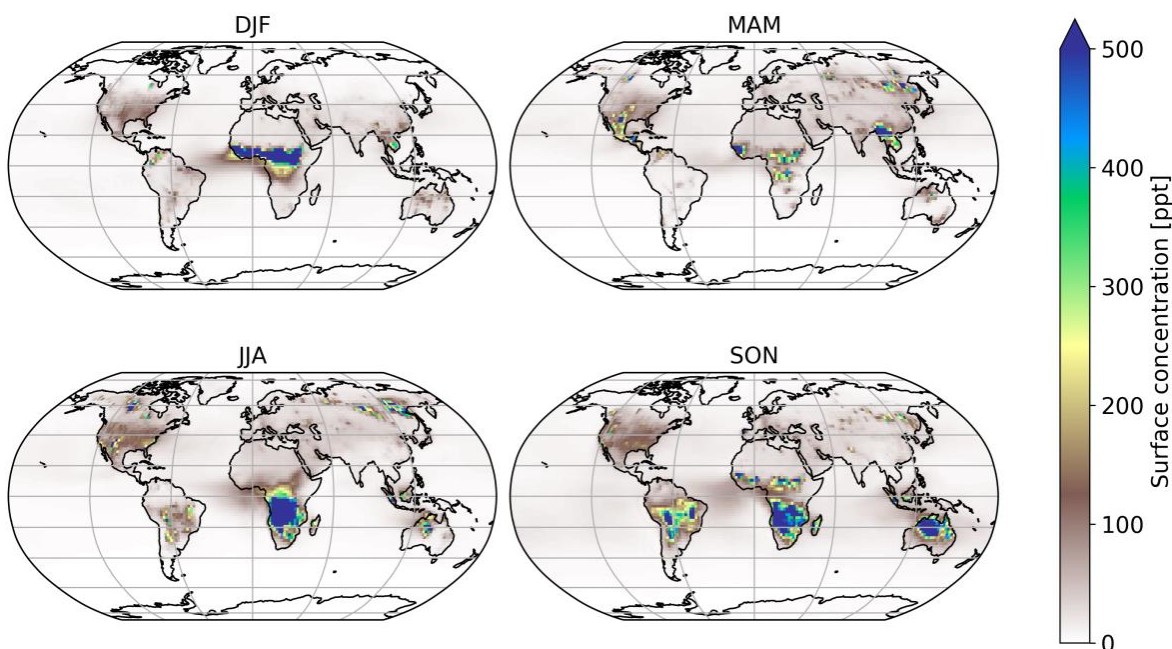

**Figure 4: Mean seasonal surface concentration of HNCO using Koss et al. (2018) biomass burning emission factors.**

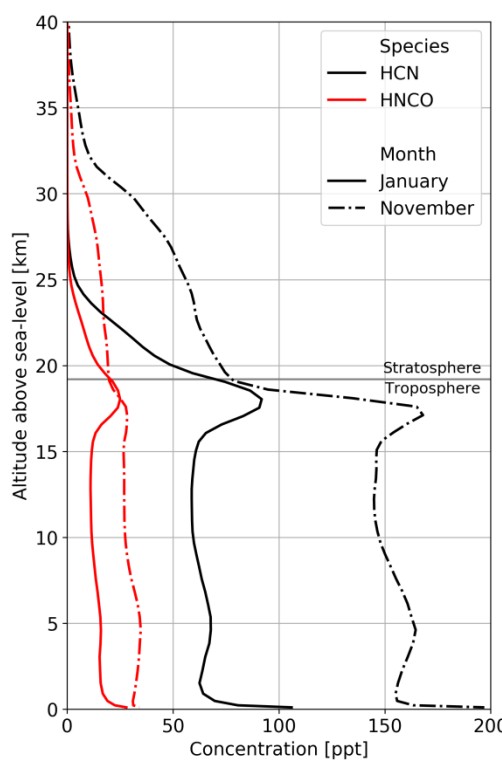

**Figure 5: Mean vertical profiles of HCN (black) and HNCO (red) for January (solid lines) and November (dash-dotted lines) over South East Asia. Biomass burning emission factors are based on Koss et al. (2018)**

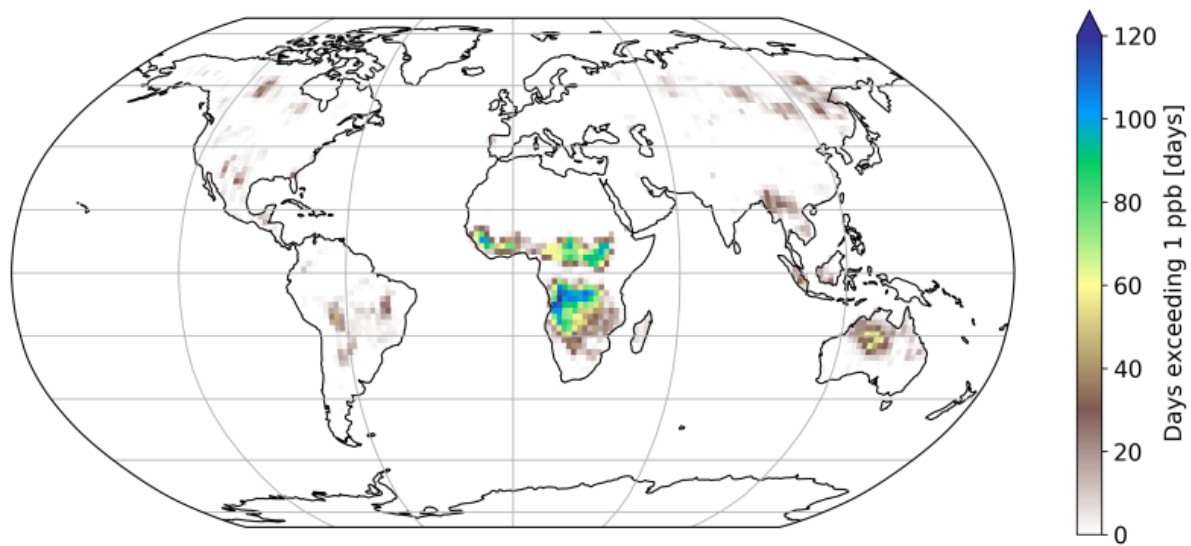

5  **Figure 6: Number of days exceeding 1 ppb of HNCO at the surface. Biomass burning emission factors are based on**
6  **Koss et al. (2018)**

