# Peer review of "Atmospheric chemical loss processes of isocyanic acid"

_Atmospheric Chemistry and Physics, 2019_

## Referee Comment (RC1) · Anonymous Referee #1 · 6 Mar 2020

This is a well-written paper with two main parts, (i) computational characterization of HNCO reactions with OH (a refinement which confirms what is known already that that this is very slow) and new data for Cl, NO3 and O3, and (ii) modeling of predicted surface concentrations of HNCO based on the results combined with literature information. My focus will be on the first part.

The dominant source of HNCO is biomass burning, and it has a long lifetime in the atmosphere, so this is a relevant topic for Atm. Chem. Phys. Fairly standard and usually reliable computational chemistry techniques are applied to characterize reactants, intermediates, products and barriers for a variety of HNCO reaction pathways. The methodology is described in sufficient detail, along with information such as molecular geometries, to allow others to reproduce the results. I see no errors in the work and

[Figure]

Creative Commons CC BY license logo

the conclusions are sound. The ms. is suitable for publication once some areas are addressed.

1. Page 3, lines 7-13. In this brief discussion of photochemistry, given the various known bond strengths in HNCO, what are the threshold wavelengths at which photodissociation could occur? 2. The uncertainty in barrier heights of 0.5 kcal/mol is quite small. How was this estimated? What are the magnitudes of the room temperature tunneling factors? I would expect large factors to be less reliable. 3. Page 6. The agreement on Fig. 2, at elevated temperatures of the order of 1000 K, is somewhat fortuitous because the slopes of computed and observed rate constants are different, but the agreement is noted favorably. Therefore factors like hindered internal rotations in the TS do need to be taken into account, even if they are not very important under atmospheric conditions. 4. Page 6 line 30 and the following line. Here and elsewhere the lifetime is described as "several decades" or 50 years. But $10^{10}$ s is about 300 years. 5. In the discussion of Cl + HNCO on p. 7 the dominant path is addition. I imagine the calculations are for the high-pressure limit, but is this reached everywhere over 200-450 K? The pressure dependence should be investigated via RRKM theory, especially as data are provided at up to 450 K where falloff is more likely than at room temperature. 6. On Line 1 of page 8 redissociation of HNC(Cl)O is stated to be its most likely fate, but given the rate quoted is the addition of molecular oxygen potentially competitive, especially below room temperature? 7. On page 8 there is no mention of the complications of describing the vibronic structure of $NO_3$ accurately with the kinds of computational approaches used here. As studied, for example, by Okumura and Stanton, Jahn-Teller effects break symmetry and change the vibrational energy levels significantly, so that standard evaluation of partition functions may be significantly in error.

Minor typographical errors: Page 3 line 3 "...data are available..." Page 6 line 26. "an" should be "a" and there is a double comma Page 10 line 2 insert a space in "whereasthe"

---

## Referee Comment (RC2) · Anonymous Referee #2 · 9 Mar 2020

General comments:

The authors describe computational chemistry calculations of HNCO with major atmospheric oxidants, including OH, NO3, CI, and O3, using reliable methods, specifically CCSD(T)/CBS(DTQ)//M06-2X/aug-cc-pVTZ. The authors' conclusion corroborates with previously known conclusions, that the gas phase chemistry is not important for HNCO's lifetime. Then, the authors use the ECHAM/MESSy Atmospheric Chemistry (EMAC) model to evaluate global concentrations of HNCO from update biomass burning inventories. The modelling effort is particularly useful for understanding which populations may be at risk of exposure. I commend the authors on this work, and on updating the model with the latest sources of HNCO and rate constants. This modelling effort is valuble to the community and appropriate to Atmos. Chem. Phys..

The work is done with reliable methods, and the conclusions, although not novel, are robust. I appreciate the authors' systematic approach of considering all possible sites of reactions even if highly improbable (for example, page 7, lines 9-10). A criticism I have is a lack of depth in the interpretation of the results (I've highlighted some specific examples below). There is also a lack of synergy between the computational data and the model; these 2 studies seem to be separated and the authors can improve their manuscript by clarifying the importance of reporting these two methods together. Was any data used in the model coming from the computational chemistry relative energies?

My second criticism is the revisiting of the gas phase reactions of an electrophile (HNCO) with other electrophilic oxidants. The tone of the manuscript suggests that the authors were surprised by this finding (for example the text on page 7 lines 1-4), when in reality is makes sense (and was known) that these oxidants would not be important for the fate of HNCO. I'm curious to know which hypothesis the authors were testing with their gas phase mechanism computational chemistry study and why they sought to do these calculations (other than it hadn't been done before).

In general, I would recommend that the authors justify their choices of methods more clearly, to help make the methods more accessible. Although it's a strength of the work to have combined computational chemistry and atmospheric chemistry modelling, it is also not typical that one has experience with both these methods. To improve the paper, a clearer description of the methods as well as a comparison with previous works would be highly beneficial for the reader. For example, how does the authors' work compare with computational methods with other techniques (basis sets) used in atmospheric chemistry such as (Møller et al., 2016; da Silva, 2013).

Finally, I would also recommend to the authors to add quantitative data throughout discussion of the manuscript by reporting values when possible. It's always best in scientific communication to be as precise as possible (specific examples are given below).
Specific comments:

Title:

The title can be better representative of the work. First, the mention of the gas phase loss processes would be very important. It should also be highlighted that the work is theoretical and modelling. A title along the lines of, "Gas phase chemical losses processes of isocyanic acid (HNCO) investigated through computational chemistry and chemical fate transport modeling" would better represent the work. One could also consider highlighting the sources of HNCO investigated, ie. biomass burning.

**Abstract:**

There is a missing introduction in the abstract to the significance of HNCO. The authors should add three to five sentences stating the research problem, then the research gap and finally their methods and findings.

I would also encourage the authors to be more specific in their abstract in order to better represent the content of the article. For example, page 1, line 19 should specify which global model is being used.

I would also encourage the authors to end their abstract with a summarizing sentence and an outlook.

Introduction:

Page 1, line 28: Why is the Wentzell reference used here? There were other references prior to this work identifying HNCO in the atmosphere. See (Roberts et al., 2010; Veres et al., 2010).

Page 1, line 31: the monomer is presentative of what? A structure? A conformer? And isomer? Or did the authors mean to write "representation of the chemical structure"?

Page 1, lines 29-32: In general, these sentences are very vague. Which concentrations, which impurities? Can the authors quantify "fairly stable" with numbers and
chemical and physical properties?

Page 1, line 33: it would be important here to specify which type of modeling study (model, year, sources) was done by Young et al. The Young et al. study is an important precursor the authors' work and more emphasis should be given to comparing this study throughout the manuscript.

Page 2, line 1: "highly toxic" is not a claim one can make based on the uncertainty between the current medical literature and a lack of exposure studies. If the authors note that 1ppbv is potentially toxic, they can explain that this number is based on thermodynamic partitioning and is not a number from empirical studies. I would revise this statement in the text.

Page 2, lines 13-15: for an additional reference for the oxidation of nicotine as a source of HNCO, see (Borduas et al., 2016a). The Hems et al. reference should be solely for cigarette smoke. The oxidation of amide references should be (Barnes et al., 2010; Borduas et al., 2015; Bunkan et al., 2015). It would be interesting for the authors to highlight the relevance of their calculations for indoor air vs outdoor air chemistry.

Page 2, line 16: The Leslie et al. review is referenced here, but best to reference the specific studies looking at these materials. See (Jankowski et al., 2014, 2016, 2017)

Page 2, line 22: these references should either be solely the review, or each study should be described. Also consider looking at (Wren et al., 2018).

Page 2, line 30: remove the word "very". I would argue that we have a good understanding of the gas phase fate of HNCO, corroborated by this manuscript.

Page 2, line 34: I think it's worth explaining in one or two sentences why the current data is only from elevated temperatures. Isn't because these reactions are all negligible at room temperature?

Page 3, lines 1-3: the authors state that there currently exists no computational data on the reactivity of NO3, CI and O3 with HNCO. Although this statement is true, why would
one expect these oxidants to react or to be important for the fate of HNCO? Based on our current knowledge of the electrophilic nature of the carbon atom in HNCO, none of these oxidants would favorably react with HNCO. Can the authors state their hypotheses (similarly to comment in the general section)?

Page 3, line 3: specify which co-reactants

Page 3, lines 4-5: the authors correctly identify the lack of knowledge and of measurements of the dry deposition of HNCO. An extended discussion on this topic is perhaps warranted. Can the authors address this important piece of missing information with their work?

Page 3, lines 7-14: the discussion on photolysis needs to be re-considered. Isn't the reason why HNCO' absorbance is only reported below 262 nm? Isn't because it doesn't absorb at higher wavelength? By this definition, one would not expect photolysis of HNCO to occur in the troposphere. The way the text is currently written suggests missing information...

Page 3, line 18-19: could be worth adding the values of KH here.

Page 3, lines 21-22: The most up-to-date calculations are arguably from (Borduas et al., 2016c; Roberts and Liu, 2019).

Methods:

Page 3, line 36: can the authors further justify their choice of basis set?

Page 4, lines 7-8: how was the tunneling correction of 1.5 calculated (perhaps also add a reference)?

Page 4, lines 15-16: I'm curious about the authors' reasoning for doing calculations despite all these reaction channels being negligible at room temperature. This point goes along with my point above which hypothesis were the authors starting with.

Page 4, paragraph on global modelling: a hierarchal diagram of the model would be

**ACPD**
useful for visualization and interpretation of the components and subcomponents of the model. This figure could go either in the main text or in the supplementary information.

Page 4, lines 25-26: what is the implication of these grid sizes on the interpretation of the results?

Page 4, line 29: why were aromatics and terpenes excluded?

Page 4, lines 31-32: have the authors considered using SAR factors for amines and amides? (see (Borduas et al., 2016b))

Page 4, lines 37-38: this conclusion would certainly depend on the scale of the modelling correct? As cigarette smoke and cooking on a very local scale could also rival biomass burning, or am I wrong?

Page 4, line 38: specify the two emission factors.

Page 5, line 2: In light of (Carter et al., 2020)'s conclusions, could the authors comment on the uncertainty of their inventory.

Page 5, line 10: the chosen years are 2010-2011. Can this choice be justified? (Young et al., 2012) used 2008 fire emissions. Can these two years be compared? Why or why not?

Results

Figure 1: Overall, this figure is particularly well done and clear. I congratulate the authors here! The structures are also done well, using ChemDraw. Actually, could all the structures be drawn with bonds and bond angles similarly to the products with ozone? In addition, where are the energies of the pre-complexes?

Page 6, line 14: could the authors show graphically on Figure 2 the mentioned experimental uncertainty?

Page 6, line 15: to be more accurate, please give the range instead of an average

**ACPD**
factor.

Page 6, line 18: specify the kinetic model.

Page 6, lines 26-27: Good conclusion, I would highlight this statement better in the abstract for example.

Page 7, line 15-16: define why an acyl chloride is less "stable" than a carboxylic acid? The current statement is rather vague.

Page 7, lines 17-18: I think this statement is wrong. Isn't it also true for HNCO + O3 based on Figure 1? It is also inconsistent with conclusions on page 8, lines 1-3

Page 8, line 6: the CI concentration is incorrect. See (Riedel et al., 2012) for example (concentrations of 10s to 100s ppt level). A ratio of OH/CI of 200 appears to be typical in urban regions based on (Young et al., 2014).

Page 8, line 17: explain what is meant by "greater stability" of nitric acid.

Page 8, lines 26-27: doesn't the sentence on lines 27-28 contradict the preceeding statement?

Page 8, line 37: specify which atoms have the reported bond energies. O-O or H-O?

Page 9, line 16: specify which mechanisms are "the described mechanisms". There are many mechanisms reported in this work.

Page 9, line 18: be consistent with writing out the name of isocyanic acid and using the molecular formula HNCO.

Page 9, line 21-22: would be interesting to show this result/data. How was this number obtained?

Page 9, lines 33-34: it would be very important to expand this sentence into a whole paragraph for comparison. And if the argument on lines 34-35 are true, then which contribution (percentage for example) is formamide a source of HNCO? This value
could be very important for the gas phase atmospheric community.

Page 9, lines 36-37: specify which heterogeneous loss terms.

Page 10, lines 1-2: this lifetime is calculated based on which values?

Page 10, lines 5-7: interesting calculation. How do the authors interpret that number with the observed diurnal profiles in (Roberts et al., 2014)?

Page 10, line 7: did the authors consider photolysis as a sink in the stratosphere? Refer to the discussion on photolysis on page 3.

Page 10, lines 12-14: show this result graphically. It is particularly interesting.

Page 10, lines 25-27: What is the role of the model's resolution in this analysis?

Page 11: useful graphic and table. Could the authors also add a column to their table relating to their own results? What is the implications of modelling different years of fire inventories when comparing the results in Table 1? I would also be consistent with significant figures throughout the table.

Page 11, line 8: is the wrong reference used here? Should it be Kumar et al 2018?

Page 12, starting at line 5: I would move this section before the global modelling data to help with the flow of the manuscript.

Page 12, line 11: describe RAPRNOx

Page 12, lines 21-22: unclear statement. What is being referred to as "total rate coefficients"?

Page 12, lines 28-29: why not be consistent with p6, line 5 with 298K?

Conclusions:

Page 13, lines 18-19: 5 Gg/y out of (insert number of total losses).

Page 13, line 19: first time ammonia is mentioned in the conclusion - the authors
can refer to (Leslie et al., 2019) for a back of the envelop calculation on ammonia budget from HNCO. How was ammonia implemented into the model? This discussion is rather important for the fate of HNCO and I would encourage the authors to discuss these numbers in the text as well.

Page 13, lines 28-30: this sentence appears to be out of place, and outside the scope of this work. Unless a connection with the authors' HNCO modeling can be made?

Technical comments:

Reference of SUVA should have the acronym written out.

Page 1, line 28: what is meant by "first recognized"? First studied? First synthesis? I would encourage the authors to be more precise.

Page 2, line 21: instead of "slippage" did the authors means "seepage" (especially for gas seeping out)?

Page 2, line 36: remove the word "very"

Page 3, line 5: bets to remove the word "recent".

Strangely enough, the numbers reported in the text do not match the numbers in the figure. Could the authors double check the numbers on: Page 5, line 24; page 5, line 26 (2 numbers); page 7, line 12.

Page 9, line 23: should be written HNCO (not HCNO).

Page 13, line 15: remove the word "fairly".

References: Barnes, I., Solignac, G., Mellouki, A. and Becker, K. H.: Aspects of the atmospheric chemistry of amides, ChemPhysChem, 11(18), 3844–3857, 2010. Borduas, N., da Silva, G., Murphy, J. G. and Abbatt, J. P. D.: Experimental and theoretical understanding of the gas phase oxidation of atmospheric amides with OH radicals: kinetics, products, and mechanisms, J. Phys. Chem. A, 119(19), 4298–
4308, doi:10.1021/jp503759f, 2015. Borduas, N., Murphy, J. G., Wang, C., da Silva, G. and Abbatt, J. P. D.: Gas Phase Oxidation of Nicotine by OH Radicals: Kinetics, Mechanisms, and Formation of HNCO, Environ. Sci. Technol. Lett., 3(9), 327-331, doi:10.1021/acs.estlett.6b00231, 2016a. Borduas, N., Abbatt, J. P. D., Murphy, J. G., So, S. and da Silva, G.: Gas-Phase Mechanisms of the Reactions of Reduced Organic Nitrogen Compounds with OH Radicals, Environ. Sci. Technol., 50(21), 11723-11734, doi:10.1021/acs.est.6b03797, 2016b. Borduas, N., Place, B., Wentworth, G. R., Abbatt, J. P. D. and Murphy, J. G.: Solubility and reactivity of HNCO in water: insights into HNCO's fate in the atmosphere, Atmospheric Chem. Phys., 16(2), 703-714, doi:10.5194/acp-16-703-2016, 2016c. Bunkan, A. J. C., Hetzler, J., Mikoviny, T., Wisthaler, A., Nielsen, C. J. and Olzmann, M.: The reactions of N-methylformamide and N,N-dimethylformamide with OH and their photo-oxidation under atmospheric conditions: Experimental and theoretical studies, Phys. Chem. Chem. Phys., 17(10), 7046-7059, doi:10.1039/C4CP05805D, 2015. Carter, T. S., Heald, C. L., Jimenez, J. L., Campuzano-Jost, P., Kondo, Y., Moteki, N., Schwarz, J. P., Wiedinmyer, C., Darmenov, A. S., Silva, A. M. da and Kaiser, J. W.: How emissions uncertainty influences the distribution and radiative impacts of smoke from fires in North America, Atmospheric Chem. Phys., 20(4), 2073–2097, doi:https://doi.org/10.5194/acp-20-2073-2020, 2020. Jankowski, M. J., Olsen, R., Nielsen, C. J., Thomassen, Y. and Molander, P.: The applicability of proton transfer reaction-mass spectrometry (PTR-MS) for determination of isocyanic acid (ICA) in work room atmospheres, Environ. Sci. Process. Impacts, 16(10), 2423-2431, doi:10.1039/C4EM00363B, 2014. Jankowski, M. J., Olsen, R., Thomassen, Y. and Molander, P.: The stability and generation pattern of thermally formed isocyanic acid (ICA) in air - potential and limitations of proton transfer reaction-mass spectrometry (PTR-MS) for real-time workroom atmosphere measurements, Environ. Sci. Process. Impacts, 18(7), 810-818, doi:10.1039/C6EM00312E, 2016. Jankowski, M. J., Olsen, R., Thomassen, Y. and Molander, P.: Comparison of air samplers for determination of isocyanic acid and applicability for work environment exposure assessment, Environ. Sci. Process. Impacts, 19(8), 1075–1085,
doi:10.1039/C7EM00174F, 2017. Leslie, M. D., Ridoli, M., Murphy, J. G. and Borduas-Dedekind, N.: Isocyanic acid (HNCO) and its fate in the atmosphere: a review, Environ. Sci. Process. Impacts, 21(5), 793-808, doi:10.1039/C9EM00003H, 2019. Møller, K. H., Otkjær, R. V., Hyttinen, N., Kurtén, T. and Kjaergaard, H. G.: Cost-Effective Implementation of Multiconformer Transition State Theory for Peroxy Radical Hydrogen Shift Reactions, J. Phys. Chem. A, 120(51), 10072–10087, doi:10.1021/acs.jpca.6b09370, 2016. Riedel, T. P., Bertram, T. H., Crisp, T. A., Williams, E. J., Lerner, B. M., Vlasenko, A., Li, S.-M., Gilman, J., de Gouw, J., Bon, D. M., Wagner, N. L., Brown, S. S. and Thornton, J. A.: Nitryl Chloride and Molecular Chlorine in the Coastal Marine Boundary Layer, Environ. Sci. Technol., 46(19), 10463–10470, doi:10.1021/es204632r, 2012. Roberts, J. M. and Liu, Y.: Solubility and solution-phase chemistry of isocyanic acid, methyl isocyanate, and cyanogen halides, Atmospheric Chem. Phys., 19(7), 4419-4437, doi:https://doi.org/10.5194/acp-19-4419-2019, 2019. Roberts, J. M., Veres, P., Warneke, C., Neuman, J. A., Washenfelder, R. A., Brown, S. S., Baasandori, M., Burkholder, J. B., Burling, I. R., Johnson, T. J., Yokelson, R. J. and de Gouw, J.: Measurement of HONO, HNCO, and other inorganic acids by negative-ion proton-transfer chemical-ionization mass spectrometry (NI-PT-CIMS): application to biomass burning emissions, Atmos.Meas.Tech., 3(4), 981–990, doi:10.5194/amt-3-981-2010, 2010. Roberts, J. M., Veres, P. R., VandenBoer, T. C., Warneke, C., Graus, M., Williams, E. J., Lefer, B., Brock, C. A., Bahreini, R., Öztürk, F., Middlebrook, A. M., Wagner, N. L., Dubé, W. P. and de Gouw, J. A.: New insights into atmospheric sources and sinks of isocyanic acid, HNCO, from recent urban and regional observations, J. Geophys. Res. Atmospheres, 119(Journal Article), doi:10.1002/2013JD019931, 2014. da Silva, G.: G3X-K theory: A composite theoretical method for thermochemical kinetics, Chem. Phys. Lett., 558(0), 109–113, doi:10.1016/j.cplett.2012.12.045, 2013. Veres, P., Roberts, J. M., Burling, I. R., Warneke, C., de Gouw, J. and Yokelson, R. J.: Measurements of gas-phase inorganic and organic acids from biomass fires by negativeion proton-transfer chemical-ionization mass spectrometry, J. Geophys. Res. Atmospheres, 115(Journal Article), D23302, doi:10.1029/2010JD014033, 2010. Wren, S.
N., Liggio, J., Han, Y., Hayden, K., Lu, G., Mihele, C. M., Mittermeier, R. L., Stroud, C., Wentzell, J. J. B. and Brook, J. R.: Elucidating real-world vehicle emission factors from mobile measurements over a large metropolitan region: a focus on isocyanic acid, hydrogen cyanide, and black carbon, Atmospheric Chem. Phys., 18(23), 16979-17001, doi:https://doi.org/10.5194/acp-18-16979-2018, 2018. Young, C. J., Washenfelder, R. A., Edwards, P. M., Parrish, D. D., Gilman, J. B., Kuster, W. C., Mielke, L. H., Osthoff, H. D., Tsai, C., Pikelnaya, O., Stutz, J., Veres, P. R., Roberts, J. M., Griffith, S., Dusanter, S., Stevens, P. S., Flynn, J., Grossberg, N., Lefer, B., Holloway, J. S., Peischl, J., Ryerson, T. B., Atlas, E. L., Blake, D. R. and Brown, S. S.: Chlorine as a primary radical: evaluation of methods to understand its role in initiation of oxidative cycles, Atmospheric Chem. Phys., 14(7), 3427–3440, doi:https://doi.org/10.5194/acp-14-3427-2014, 2014. Young, P. J., Emmons, L. K., Roberts, J. M., Lamargue, J.-F., Wiedinmyer, C., Veres, P. and VandenBoer, T. C.: Isocyanic acid in a global chemistry transport model: tropospheric distribution, budget, and identification of regions with potential health impacts, J. Geophys. Res. Atmospheres, 117(Journal Article), D10308, doi:10.1029/2011JD017393, 2012.

---

## Author Comment (AC1) · 20 Apr 2020

**Answer to referee comments**

We would like to thank the referees for their extensive comments, which helped significantly to improve the manuscript in content and readability.

**Anonymous Referee #1** This is a well-written paper with two main parts, (i) computational characterization of HNCO reactions with OH (a refinement which confirms what is known already that that this is very slow) and new data for Cl, NO3 and O3, and (ii) modeling of predicted surface concentrations of HNCO based on the results combined with literature information. My focus will be on the first part. The dominant source of HNCO is biomass burning, and it has a long lifetime in the atmosphere, so this is a relevant topic for Atm. Chem. Phys. Fairly standard and usually reliable computational chemistry techniques are applied to characterize reactants, intermediates, products and barriers for a variety of HNCO reaction pathways. The methodology is described in sufficient detail, along with information such as molecular geometries, to allow others to reproduce the results. I see no errors in the work and the conclusions are sound. The ms. is suitable for publication once some areas are addressed.
1. Page 3, lines 7-13. In this brief discussion of photochemistry, given the various known bond strengths in HNCO, what are the threshold wavelengths at which photodissociation could occur?

The threshold for HNCO dissociation limit starts at wavelengths below 240 nm forming either H+NCO or NH + CO; this is now mentioned and referenced in the text. We also refer to a study at lower energies, but the lifetime would be too long to make it relevant, so we conclude that photolysis is not important. We did not calculate the bond dissociation energies to compare against the experimental thresholds, as these would be calculated for the ground state whereas the photolysis occurs predominantly through the first singlet exited states. Calculating the vertical excitation energies to excited states is well beyond the scope of the present paper, and would require different theoretical methodologies.

2. The uncertainty in barrier heights of 0.5 kcal/mol is quite small. How was this estimated? What are the magnitudes of the room temperature tunneling factors? I would expect large factors to be less reliable.

That should read "at least 0.5 kcal/mol", we apologize for the confusion. Benchmark studies tend to examine overall uncertainty and thus include many different reaction classes with a wide variety of reactants/molecules, and the uncertainty estimate is thus influenced unduly by reaction classes well beyond the more standard H-abstraction or addition in organic atmospheric chemistry as studied here. Still, CCSD(T)/CBS has been referred to as the "golden standard" because it typically provides chemical accuracy or better. Nowadays, (composite) post-CCSD(T) methods can be used to obtain sub-chemical accuracy with only a few tenths of kcal/mol of uncertainty.
The total factor of 4 on the rate coefficient is estimated from our experience where we typically observe a difference smaller than that when compared against experiment, in many different reaction classes.

3. Page 6. The agreement on Fig. 2, at elevated temperatures of the order of 1000 K, is somewhat fortuitous because the slopes of computed and observed rate constants are different, but the agreement is noted favorably. Therefore factors like hindered internal rotations in the TS do need to be taken into account, even if they are not very important under atmospheric conditions.
5. In the discussion of Cl + HNCO on p. 7 the dominant path is addition. I imagine the calculations are for the high-pressure limit, but is this reached everywhere over 200-450 K? The pressure dependence should be investigated via RRKM theory, especially as data are provided at up to 450 K where falloff is more likely than at room temperature
7. On page 8 there is no mention of the complications of describing the vibronic structure of NO3 accurately with the kinds of computational approaches used here. As studied, for example, by Okumura and Stanton, Jahn-Teller effects break symmetry and change the vibrational energy levels significantly, so that standard evaluation of partition functions may be significantly in error.

While we feel that our calculations are of a high quality, we acknowledge that there are some aspects that could be improved, if one were interested in doing benchmark-level calculations. For the current purposes, i.e. documenting their (lack of) impact on atmospheric chemistry, there is a strongly diminished return in implementing these computationally expensive improvements, for changes that are likely less than a factor of 2, irrelevant compared to the orders of magnitude in rate separating gas phase loss processes from the dominant atmospheric sinks. As such,

we choose not to dedicate the necessary resources. We have included an additional paragraph in the methodology section explicitly discussing these refinements of the predictions. We have also included a line in figure 2 (rate coefficient of HNCO+OH) indicating the rate coefficient that would be needed to make the OH reaction significant on a global scale, to further visualize that the conclusions are robust against minor uncertainties in the predictions.

4. Page 6 line 30 and the following line. Here and elsewhere the lifetime is described as "several decades" or 50 years. But 10^10 s is about 300years.

The given lifetime inadvertently lost its pre-exponential significand, increasing the printed lifetime by almost an order of magnitude; the original number is correct and mean lifetime is about 40-50 years. We still changed the wording to "decades to centuries" here and elsewhere when discussing the local gas phase chemical losses. The impact section also has a newly added discussion of the airparcel-specific lifetime of HNCO, where depending on the location in the atmosphere we find lifetimes from 6 years to >500 years.

6. On Line 1 of page 8 redissociation of HNC(Cl)O is stated to be its most likely fate, but given the rate quoted is the addition of molecular oxygen potentially competitive, especially below room temperature?

To our knowledge, there is no data for O2 addition to this type of N-centered, delocalized multifunctionalized radicals. For vinoxy radicals, typical first-order rate coefficients in the atmosphere are of the order of 6E6 s-1 (k(298K) ~ 1E-12 cm3 molecule-1 s-1) but for small species as here redissociation of the $RO_2$ adduct back to R + $O_2$ is likely an important path. Hence, even at 200K redissociation of HNC(Cl)O to HNCO + Cl is expected to remain the main route. We have added a sentence to this effect in the paper, with reference. Even if later data indicates that we have severely underestimated the O2 addition rate, the HNCO+Cl reaction will remain a negligible atmospheric sink.

Minor typographical errors:
Page 3 line 3 "...data are available..."
Page 6 line 26."an" should be "a" and there is a double comma
Page 10 line 2 insert a space in"whereasthe"

We corrected the typographical errors accordingly

**Anonymous Referee #2**: General comments: The authors describe computational chemistry calculations of HNCO with major atmospheric oxidants, including OH, NO3, Cl, and O3, using reliable methods, specifically CCSD(T)/CBS(DTQ)//M06-2X/aug-cc-pVTZ. The authors' conclusion corroborates with previously known conclusions, that the gas phase chemistry is not important for HNCO's lifetime. Then, the authors use the ECHAM/MESSy Atmospheric Chemistry (EMAC) model to evaluate global concentrations of HNCO from update biomass burning inventories. The modelling effort is particularly useful for understanding which populations may be at risk of exposure. I commend the authors on this work, and on updating the model with the latest sources of HNCO and rate constants. This modelling effort is valuable to the community and appropriate to Atmos. Chem. Phys. The work is done with reliable methods, and the conclusions, although not novel, are robust. I appreciate the authors' systematic approach of considering all possible sites of reactions even if highly improbable (for example, page 7, lines 9-10). A criticism I have is a lack of depth in the interpretation of the results (I've highlighted some specific examples below).

There is also a lack of synergy between the computational data and the model; these 2 studies seem to be separated and the authors can improve their manuscript by clarifying the importance of reporting these two methods together. Was any data used in the model coming from the computational chemistry relative energies?

We apologize for this aspect not being clear. We have modified the initial part of the abstract, the last paragraph of section 1 (introduction) and the beginning of section 4 (global impact), to better clarify the synergy.

My second criticism is the revisiting of the gas phase reactions of an electrophile (HNCO) with other electrophilic oxidants. The tone of the manuscript suggests that the authors were surprised by this finding (for example the text on

page 7 lines 1-4), when in reality is makes sense (and was known) that these oxidants would not be important for the fate of HNCO. I'm curious to know which hypothesis the authors were testing with their gas phase mechanism computational chemistry study and why they sought to do these calculations (other than it hadn't been done before).

We ourselves do not sense that tone of surprise in our text. Even before the first calculation was done, we expected to find reaction rates that would confirm negligible contributions. We also do not try to make them appear more important than they are, as is all too often done with negative results, but clearly state where necessary that the reactions can't contribute.  Since these reactions were not studied before (at room temperature, for OH), these are new results and we can't easily refer to literature data to fully underline our lack of surprise.
We studied these reactions for a number of reasons. Initially, it was out of mild curiosity how correct the linear extrapolation of the Arrhenius expression from the experimental data for HNCO + OH to room temperature would be, as tunnelling would lead to a curved Arrhenius plot and hence a higher rate coefficient. We indeed found this curvature, but as the TS is broad and low, tunnelling is not all that important, and the curvature is too limited to make a critical difference.
For the other reactions, it is indeed "expected" that the reactions would be slow, but not "known" as no direct data was available. The molecules are fairly small, the computational burden, even at high levels of methodology, is not too high, so we characterized their entrance channels.
With 4 different atomic types in a single molecule, the subsequent chemistry is diverse, an interesting teaching case for early-stage computational scientists, and again the calculations are not overly taxing on resources, so for some of the reactions the extended PES was also investigated (see supporting information). These PESs, other than a brief summary of NCO chemistry, was kept strictly in the supporting information as it is of little to no use for the target audience of this paper.
So, by and large, there was no grand research plan for the theoretical data that led to these negative theoretical results for OH/Cl/NO3/O3, but rather academic interest in filling in some gaps in knowledge that mushroomed a bit beyond the original topic.

In general, I would recommend that the authors justify their choices of methods more clearly, to help make the methods more accessible. Although it's a strength of the work to have combined computational chemistry and atmospheric chemistry modelling, it is also not typical that one has experience with both these methods. To improve the paper, a clearer description of the methods as well as a comparison with previous works would be highly beneficial for the reader. For example, how does the authors' work compare with computational methods with other techniques (basis sets) used in atmospheric chemistry such as (Møller et al., 2016; da Silva, 2013).

We now refer to Vereecken and Francisco, 2012, and Vereecken et al. 2017, for some reviews on the relative merits of theoretical methodologies used in atmospheric chemistry, and the Papajak and Truhlar 2012 paper for basis set choice.
The relative benefits of the myriad of experimental methods, modelling methodologies, quantum chemical methodologies, and theoretical methodologies can't be discussed in each paper, not even concisely, as there are simply too many aspects to discuss. This is even more true when trying to explain this to someone not familiar with the methodologies (as I suspect most readers of this paper won't be), as this would require a very lengthy tutorial first, to then discuss subtleties that can't possibly be covered in a tutorial. The methods used in this work are at the current computational sweet spot, where going beyond the used level would require exponentially higher computational resources, well beyond what is warranted for reactions that are expected to be negligible; at the same time we are using robust methods for all aspects that are known to nearly always provide reliable results.
The Møller et al. 2016 method is a watered-down version of our MC-TST multi-conformer method  (Vereecken and Peeters, 2003), using very low levels of theory to investigate the conformational space of molecules. The supporting information of Novelli et al. 2019 contains several pages of discussion on the differences between Møller et al 2016 and Vereecken and Peeters 2003. Neither of these is truly applicable to the current PES as the intermediates here don't have many conformers.
da Silva 2013 is an example of a composite method, of which there are hundreds defined and dozens used on regular basis (CBS-Q, CBS-QB3, G2, G3SX, G4, …). These all aim to combine computationally less expensive methods to estimate the result that would have been obtained at a higher level of theory. The method used in the current study, CCSD(T)/CBS, is often the method aimed for by these composite methods for its general reliability, and the use of composite methods is not expected to provide a better result.

Vereecken, L. and Peeters, J.: The 1,5-H-shift in 1-butoxy: A case study in the rigorous implementation of transition state theory for a multirotamer system, J. Chem. Phys., 119(10), 5159–5170, doi:10.1063/1.1597479, 2003.

Novelli, A., Vereecken, L., Bohn, B., Dorn, H.-P., Gkatzelis, G. I., Hofzumahaus, A., Holland, F., Reimer, D., Rohrer, F., Rosanka, S., Taraborrelli, D., Tillmann, R., Wegener, R., Yu, Z., Kiendler-Scharr, A., Wahner, A. and Fuchs, H.: Importance of isomerization reactions for the OH radical regeneration from the photo-oxidation of isoprene investigated in the atmospheric simulation chamber SAPHIR, Atmos. Chem. Phys., 20, 3333–3355, doi:acp-20-3333-2020, 2020.

Vereecken, L. and Francisco, J. S.: Theoretical studies of atmospheric reaction mechanisms in the troposphere, Chem. Soc. Rev., 41(19), 6259–6293, doi:10.1039/c2cs35070j, 2012.

Vereecken, L., Glowacki, D. R. and Pilling, M. J.: Theoretical Chemical Kinetics in Tropospheric Chemistry: Methodologies and Applications, Chem. Rev., 115(10), 4063–4114, doi:10.1021/cr500488p, 2015.

Papajak, E. and Truhlar, D. G.: What are the most efficient basis set strategies for correlated wave function calculations of reaction energies and barrier heights?, J. Chem. Phys., 137(6), 064110, doi:10.1063/1.4738980, 2012.

Finally, I would also recommend to the authors to add quantitative data throughout discussion of the manuscript by reporting values when possible. It's always best in scientific communication to be as precise as possible (specific examples are given below).

We have extended the text in several places by adding the specific values obtained from our work or the literature, making the manuscript much easier to follow.

Specific comments:
Title: The title can be better representative of the work. First, the mention of the gas phase loss processes would be very important. It should also be highlighted that the work is theoretical and modelling. A title along the lines of, "Gas phase chemical losses processes of isocyanic acid (HNCO) investigated through computational chemistry and chemical fate transport modeling" would better represent the work. One could also consider highlighting the sources of HNCO investigated, ie. biomass burning.

We have updated the title to "*Atmospheric chemical loss processes of isocyanic acid (HNCO): a combined theoretical kinetic and global modelling study*". Since we also cover heterogeneous loss, we chose not to add "gas-phase". As we do not make significant new contributions to the HNCO sources, but rely mostly on pre-existing emissions, this was also not included in the title.

Abstract: There is a missing introduction in the abstract to the significance of HNCO. The authors should add three to five sentences stating the research problem, then the research gap and finally their methods and findings. I would also encourage the authors to be more specific in their abstract in order to better represent the content of the article. For example, page 1, line 19 should specify which global model is being used. I would also encourage the authors to end their abstract with a summarizing sentence and an outlook.

The abstract was rephrased and extended to include this information.

Introduction: Page 1, line 28: Why is the Wentzell reference used here? There were other references prior to this work identifying HNCO in the atmosphere. See (Roberts et al., 2010; Vereset al., 2010).

The used references (Roberts et al., 2011;Wentzell et al., 2013) are based on measurements in the ambient air. Therefore, these are appropriate for this position. The suggested references (Roberts et al., 2010; Veres et al., 2010) are mostly related to lab measurements. The reference (Wentzell et al., 2013) is only used to give a perspective of the presence of HNCO in the urban ambient air other than the USA.

Page 1, line 31: the monomer is presentative of what? A structure? A conformer? And isomer? Or did the authors mean to write "representation of the chemical structure"?

We rephrased to state that HNCO is near-exclusively present as the monomer in the gas phase at ambient conditions.

Page 1, lines 29-32: In general, these sentences are very vague. Which concentrations, which impurities? Can the authors quantify "fairly stable" with numbers and chemical and physical properties?

These sentences are modified to make the text more readable and avoid vagueries, with mention of the concentration level and the discussion is mainly kept to gaseous-phase.

Page 1, line 33: it would be important here to specify which type of modeling study(model, year, sources) was done by Young et al. The Young et al. study is an important precursor the authors' work and more emphasis should be given to comparing this study throughout the manuscript.

The sentence is modified and more information is included

Page 2, line 1: "highly toxic" is not a claim one can make based on the uncertainty between the current medical literature and a lack of exposure studies. If the authors note that 1ppbv is potentially toxic, they can explain that this number is based on thermodynamic partitioning and is not a number from empirical studies. I would revise this statement in the text.

The text is revised accordingly. The concentration is now referred as "estimated". Details about the calculation are not provided as it would be a lengthy addition and it has been discussed extensively by other studies (Roberts et al., 2011; Leslie et al., 2019).

Page 2, lines 13-15: for an additional reference for the oxidation of nicotine as a source of HNCO, see (Borduas et al., 2016a). The Hems et al. reference should be solely for cigarette smoke. The oxidation of amide references should be (Barnes et al., 2010;Borduas et al., 2015; Bunkan et al., 2015). It would be interesting for the authors to highlight the relevance of their calculations for indoor air vs outdoor air chemistry.

The references are modified accordingly. Here we mainly wish to highlight the presence of HNCO in cigarette smoke (inhaled). Our paper focuses mostly on global models, and a discussion of indoor/outdoor chemistry is outside the scope of the paper.

Page 2, line 16: The Leslie et al. review is referenced here, but best to reference the specific studies looking at these materials. See (Jankowski et al., 2014, 2016, 2017)

The references are modified accordingly.

Page 2, line 22: these references should either be solely the review, or each study should be described. Also consider looking at (Wren et al., 2018).

The references are modified accordingly.

Page 2, line 30: remove the word "very". I would argue that we have a good under-standing of the gas phase fate of HNCO, corroborated by this manuscript.

The word "very" is removed. The statement referred to the total lack of data for NO3/Cl/O3, and the absence of room temperature data for OH; the second part of the sentence already concurs with the referee that sufficient indirect data exists to infer that these reactions are slow.

Page 2, line 34: I think it's worth explaining in one or two sentences why the current data is only from elevated temperatures. Isn't because these reactions are all negligible at room temperature?

It is mostly because the reactions were studied for combustion research, with HNCO being a critical intermediate in chemical NOx reduction strategies (a hot topic at the end of last century but since then mostly obsoleted by catalytic reduction flue gas treatments), and thus using experimental setups appropriate for those conditions. In my younger years I (LV) even published a theoretical study on HNCO + H (not cited as not relevant). The consideration of HNCO at atmospheric conditions is, in comparison, fairly recent, prompted by its emission from AdBlue and similar

SCR flue gas treatments (only deployed in quantity in the past decade), and increased interest in the atmospheric impact of biomass burning and wildfires. We added a short note linking the experimental data to the relevant research setting.

Page 3, lines 1-3: the authors state that there currently exists no computational data on the reactivity of NO3, Cl and O3 with HNCO. Although this statement is true, why would one expect these oxidants to react or to be important for the fate of HNCO? Based on our current knowledge of the electrophilic nature of the carbon atom in HNCO, none of these oxidants would favorably react with HNCO. Can the authors state their hypotheses (similarly to comment in the general section)?

See our earlier answer with regard to the reasons for doing this study.

Page 3, line 3: specify which co-reactants

The coreactants O, H, CN, HCO have been added to the text

Page 3, lines 4-5: the authors correctly identify the lack of knowledge and of measurements of the dry deposition of HNCO. An extended discussion on this topic is perhaps warranted. Can the authors address this important piece of missing information with their work?

It is difficult to address this missing information with the current study. However, a separate discussion on this matter and comparisons to Young et al., 2012 is desirable. We thus included a description in the methodology section of the global model of how dry deposition was modelled. A discussion on dry deposition was added to the manuscript and additional information on dry deposition was added to Table 1.

Page 3, lines 7-14: the discussion on photolysis needs to be reconsidered. Isn't the reason why HNCO' absorbance is only reported below 262 nm? Isn't because it doesn't absorb at higher wavelength? By this definition, one would not expect photolysis of HNCO to occur in the troposphere. The way the text is currently written suggests missing information...

We have expanded and rephrased our discussion of the photolysis, with additional references, and now explicitly state that photolysis appears to be of no importance in the lower atmosphere.

Page 3, line 18-19: could be worth adding the values of KH here.

Added

Page 3, lines 21-22: The most up-to-date calculations are arguably from (Borduas etal., 2016c; Roberts and Liu, 2019).

We now refer to the most recent study by Roberts and Liu, 2019

Methods:
Page 3, line 36: can the authors further justify their choice of basis set?

See our earlier comment on the relative merits of the chosen methodologies, and the references listed there.

Page 4, lines 7-8: how was the tunneling correction of 1.5 calculated (perhaps also add a reference)?

Calculation of the tunnelling correction was done using Eckart tunnelling corrections for asymmetric barriers. This is a very widely used, and comparatively simple methodology, that is referenced in the methodology.

Page 4, lines 15-16: I'm curious about the authors' reasoning for doing calculations despite all these reaction channels being negligible at room temperature. This point goes along with my point above which hypothesis were the authors starting with.

See our earlier answer with regard to the reasons for doing this study.

Page 4, paragraph on global modelling: a hierarchal diagram of the model would be useful for visualization and interpretation of the components and subcomponents of the model. This figure could go either in the main text or in the supplementary information.

A model hierarchal diagram of EMAC was already provided in Jöckel et al., 2005 and an updated overview of all model components is given in Jöckel et al., 2010. Describing those technical parts of the model is out of the scope of our study, and we cannot do this justice without adding an overly long discussion in the paper. However, additional references to these papers were added to the manuscript for readers interested in these aspects..

Jöckel, P., Sander, R., Kerkweg, A., Tost, H., and Lelieveld, J.: Technical Note: The Modular Earth Submodel System (MESSy) - a new approach towards Earth System Modeling, Atmos. Chem. Phys., 5, 433–444, https://doi.org/10.5194/acp-5-433-2005, 2005.
Jöckel, P., Kerkweg, A., Pozzer, A., Sander, R., Tost, H., Riede, H., Baumgaertner, A., Gromov, S., and Kern, B.: Development cycle 2 of the Modular Earth Submodel System (MESSy2), Geosci. Model Dev., 3, 717–752, https://doi.org/10.5194/gmd-3-717-2010, 2010.

Page 4, lines 25-26: what is the implication of these grid sizes on the interpretation of the results?

Using this horizontal resolution allows us to assess the impact of HNCO chemical losses while still being computationally affordable. Therefore, we can reliably estimate the global impact, but not the regional ones (e.g. single megacities). By using 90 layers, focusing on the lower and middle atmosphere, we are able to cover the whole troposphere. Additionally, vertical transport processes are represented in a good manner (Jöckel et al., 2010) which allows us to also investigate the impact on the UTLS. An elaborate explanation was added to the text.

Jöckel, P., Kerkweg, A., Pozzer, A., Sander, R., Tost, H., Riede, H., Baumgaertner, A., Gromov, S. and Kern, B.: Development cycle 2 of the Modular Earth Submodel System (MESSy2), Geosci. Model Dev., 3(2), 717–752, doi:10.5194/gmd-3-717-2010, 2010.

Page 4, line 29: why were aromatics and terpenes excluded?

For global model studies the balance of computational demand is of importance. We consider aromatics and terpenes to be of little importance for the processes studied. Thus we excluded them to reduce the computational demand. A statement about this was added to the manuscript.

Page 4, lines 31-32: have the authors considered using SAR factors for amines and amides? (see (Borduas et al., 2016b))

The reaction rates used are average values from all reported experimental data given in Nielsen et al., 2012. The product yields used are the product yields from the same source, including some simplifications. In general, we prefer measured reaction rates over SAR estimates, though the latter are surely useful when modelling the chemical kinetics of larger amines and amides. Using additional SAR factors will likely not improve the product yields accuracy, and thus not improve the reliability of the proposed mechanism. The section covering the global model description was updated to include a more elaborated description on the reaction rates and reaction yields used.

Page 4, lines 37-38: this conclusion would certainly depend on the scale of the modelling correct? As cigarette smoke and cooking on a very local scale could also rival biomass burning, or am I wrong?

Indeed, on local scales other sources could be more significant. However, we consider these sources to be of no importance on a global scale, and due to the grid size used for this global model application such sources would not show in the predictions. Therefore, these sources are not taken into account. Modelling e.g. cigarette smoke is hardly possible within the given resolution. A short explanation was added to the text.

Page 4, line 38: specify the two emission factors.

Done

Page 5, line 2: In light of (Carter et al., 2020)'s conclusions, could the authors comment on the uncertainty of their inventory.

Using different dry matter burned values will definitely impact the primary and secondary biomass burning emissions of HNCO. Estimating the uncertainties introduced by using different dry matter burned sources in EMAC is laborious. The reason is that online calculations of emission strengths are only possible with GFAS in the model version we have used. This will be changed in the next EMAC version (v 2.55). On a global scale, dry matter burned is at the higher end in GFAS but still lower than FINN and QFED and almost twice as high as GFED (Figure 4 in Carter et al., 2020). We expect similar global HNCO burdens simulated when using FINN or QFED. When using GFED, the reduced dry matter burned would result in a lower HNCO burden.

Carter, T. S., Heald, C. L., Jimenez, J. L., Campuzano-Jost, P., Kondo, Y., Moteki, N., Schwarz, J. P., Wiedinmyer, C., Darmenov, A. S., da Silva, A. M., and Kaiser, J. W.: How emissions uncertainty influences the distribution and radiative impacts of smoke from fires in North America, Atmos. Chem. Phys., 20, 2073–2097, https://doi.org/10.5194/acp-20-2073-2020, 2020.

Page 5, line 10: the chosen years are 2010-2011. Can this choice be justified? (Younget al., 2012) used 2008 fire emissions. Can these two years be compared? Why or why not?

The model is initialised using datasets which do not include any information on HNCO (since HNCO was not implemented in EMAC so far). The time period 2010-2011 was chosen, since in 2010 a particular high fire radiative energy was observed (Figure 4 in Kaiser et al., 2012). This results in high biomass burning emissions, leading to high HNCO background concentrations in the spin-up period, which is favourable for the analysis in 2011. Based on Figure 4 in Kaiser et al., 2012, a similar situation is given in 2007. Young et al., 2012 do not provide any information on their spin-up period used. If 2007 was used, the analysis performed in Young et al., 2012 for 2008 would be comparable. A short statement on this matter was added to the manuscript.

Kaiser, J. W., Heil, A., Andreae, M. O., Benedetti, A., Chubarova, N., Jones, L., Morcrette, J.-J., Razinger, M., Schultz, M. G., Suttie, M., and van der Werf, G. R.: Biomass burning emissions estimated with a global fire assimilation system based on observed fire radiative power, Biogeosciences, 9, 527–554, https://doi.org/10.5194/bg-9-527-2012, 2012.

Results
Figure 1: Overall, this figure is particularly well done and clear. I congratulate the authors here! The structures are also done well, using ChemDraw. Actually, could all the structures be drawn with bonds and bond angles similarly to the products with ozone? In addition, where are the energies of the pre-complexes?

The supporting information already has 3D representations with bond lengths and angles for all relevant structures. To avoid overloading the main paper with theoretical data that ultimately yields rate data that is unlikely to be included in the models, we choose not to include it in the main text. For the same reason, the pre-reaction complexes are omitted as they do not affect the kinetics significantly. We now state this explicitly in the caption for figure 1, and refer to the supporting information.

Page 6, line 14: could the authors show graphically on Figure 2 the mentioned experimental uncertainty?

The uncertainty is now indicated as a shaded area. We have also added an indication to this graph which rate coefficient would be needed to make HNCO loss through OH reaction relevant for the atmosphere.

Page 6, line 15: to be more accurate, please give the range instead of an average factor.

We feel that using a range is actually less informative and more complex, as the uncertainty interval is highly asymmetric. Most uncertainties in theoretical calculations, such as on the barrier height or vibrational characteristics, are incorporated in an exponential factor in the rate calculations, and the symmetric, additive uncertainty on the input data leads to a symmetric multiplicative uncertainty readily reported as a factor. Tunneling uncertainties have

likewise non-linear response function. Reporting this as an additive uncertainty requires an asymmetric error with different upper and lower bounds, which is harder to interpret. Reporting it as a range is misleading, as it is then typically assumed that the most likely value is the middle of the interval, contrary to the actual computational result. Symmetrising the additive uncertainty interval either overstates or understates the error. We thus strongly prefer to report our uncertainties as factors.

Page 6, line 18: specify the kinetic model.

This refers to the kinetic model defined in the methodologies. We have changes this to "the theoretical kinetic calculation" and "the predictions" to avoid confusion with the unrelated atmospheric model.

Page 6, lines 26-27: Good conclusion, I would highlight this statement better in the abstract for example.

We now state explicitly both in the abstract and conclusions that the reaction occurs by H-abstraction.

Page 7, line 15-16: define why an acyl chloride is less "stable" than a carboxylic acid? The current statement is rather vague.
Page 8, line 17: explain what is meant by "greater stability" of nitric acid.

The difference reaction energies are due to molecular differences in induction, orbital overlap, electron distribution as influenced by electronegativity, H-bonding... These are also the effects that e.g. make acyl chlorides much more reactive than carboxylic acids, and hence more useful in organic synthesis. As this paper is not the appropriate place to go in detail on the molecular orbital layout of the respective adducts, the statement was removed, and we now refer only to the reduced reaction energy, assuming all readers are familiar with Bell-Evans-Polanyi relationships.

Page 7, lines 17-18: I think this statement is wrong. Isn't it also true for HNCO + O3 based on Figure 1? It is also inconsistent with conclusions on page 8, lines 1-3

As stated in the ozonolysis reaction section, H-abstraction contributes for 80% in HNCO + O3, despite not having the lowest energy barrier, in agreement with our statement that Cl is only of the studied mechanisms where addition is the main entrance channel. Stating that the Cl addition path is dominant remains consistent with the later conclusion that despite the higher reaction flux through that channel, it does not effectively lead to product formation due to another reaction (in this case adduct redissociation) undoing the product formation. This is true in in atmospheric conditions, but not true in general, and we now emphasise this more, using also the estimated O2 addition pseudo-first order rate coefficient.
We also changed our wording in several places from "the reaction occurs through H-abstraction" (or similar) to "HNCO removal occurs through H-abstraction" (or similar).

Page 8, line 6: the Cl concentration is incorrect. See (Riedel et al., 2012) for example (concentrations of 10s to 100s ppt level). A ratio of OH/Cl of 200 appears to be typical in urban regions based on (Young et al., 2014).

Reidel et al. 2012 reports concentrations of 10s to 100s ppt for chlorinated compounds but did not measure Cl-atom concentrations: these were deduced from a box model based on ClNO2 and Cl2 measurements limited to the coastal region; they make no statements on the global average Cl concentration that I could find. However, even using the momentary peak concentrations in that region of about $\sim 10^5$ Cl / cm$^{-3}$ reported in the supporting information of Reidel et al. 2012 would not make this reaction important.
Our global model simulations neglect the heterogeneous production of $ClNO_2$ and compute tropospheric chlorine radical concentrations of about $2 \times 10^3$ molecules cm$^{-3}$, comparable to the number stated in the text.
We have now included a paragraph looking at minimum and maximum co-reactant concentrations, and estimated lifetimes in the reaction conditions where these maxima are found, further strengthening the link between the theoretical predictions and the model.

Riedel, T. P., Bertram, T. H., Crisp, T. A., Williams, E. J., Lerner, B. M., Vlasenko, A., Li, S.-M., Gilman, J., de Gouw, J., Bon, D. M., Wagner, N. L., Brown, S. S. and Thornton, J. A.: Nitryl Chloride and Molecular Chlorine in the Coastal Marine Boundary Layer, Environ. Sci. Technol., 46(19), 10463–10470, doi:10.1021/es204632r, 2012.

Page 8, lines 26-27: doesn't the sentence on lines 27-28 contradict the preceding statement?

There is no contradiction: removal by NO3 at night is comparable to removal by OH by day, and both are ineffective compared to other loss processes. We now state "likewise considered" instead of "still considered" to make this more clear.

Page 8, line 37: specify which atoms have the reported bond energies. O-O or H-O?

It is stated explicitly that it falls apart to OH + O2.

Page 9, line 16: specify which mechanisms are "the described mechanisms". There are many mechanisms reported in this work.

This sentence refers explicitly to the mechanism in table 1 and 2 in the supplementary material. Changed the wording to "kinetic model".

Page 9, line 18: be consistent with writing out the name of isocyanic acid and using the molecular formula HNCO.

We have harmonized mostly on notation as HNCO, other than to mention the chemical name in the abstract, introduction and conclusions, for the benefit of readers that only skim these sections.

Page 9, line 21-22: would be interesting to show this result/data. How was this number obtained?

The number was obtained by taking the total HNCO mass above the planetary boundary layer and below the tropopause into account and compare it to the total modelled HNCO mass. A figure was added showing the vertical profile of HNCO and HCN for different areas and time periods. This figure is referred to in the text to improve this discussion.

Page 9, lines 33-34: it would be very important to expand this sentence into a whole paragraph for comparison. And if the argument on lines 34-35 are true, then which contribution (percentage for example) is formamide a source of HNCO? This value could be very important for the gas phase atmospheric community.

Formamide as a source of HNCO heavily depends on the emission factors used. Information on this matter was added to the text, and can be found in Table 1 for the simulation using the emission factors of Koss et al., 2018 and Kumar et al., 2018. Table 1 also provides a detailed comparison between our results and the results from Young et al., 2012. The table headers were changed to make it more clear that the results given in this table tabulates results from our own study, using the aforementioned emission factors.

Page 9, lines 36-37: specify which heterogeneous loss terms.

It is based on dry and wet deposition. This information was added to the text.

Page 10, lines 1-2: this lifetime is calculated based on which values?

The atmospheric lifetime is calculated based on all HNCO loss terms (chemical and heterogeneous). The chemical lifetime is based solely on the chemical losses. The chemical lifetime is significantly higher, since chemical losses are significantly lower than heterogeneous loss terms. The information on how these values are calculated is added to the text.

Page 10, lines 5-7: interesting calculation. How do the authors interpret that number with the observed diurnal profiles in (Roberts et al., 2014)?

As mentioned in the manuscript, we only obtained daily mean values in this study. Comparing these to diurnal profiles would unfortunately require to repeat expensive model simulations. Additionally, we expect the model to reproduce the diurnal profiles of surface HNCO discussed by Roberts et al., 2014 since we take the strong

secondary source from formamide into account. Such a comparison may be subject of a subsequent study that could make use of in-house HNCO measurements in urban environments and possibly in the stratosphere, as part of ongoing research.

Page 10, line 7: did the authors consider photolysis as a sink in the stratosphere? Refer to the discussion on photolysis on page 3.

Within the global modelling photolysis as a stratospheric sink was not taken into account. This information was added to the discussion.

Page 10, lines 12-14: show this result graphically. It is particularly interesting.Page 10, lines 25-27: What is the role of the model's resolution in this analysis?

An additional figure was added to show this graphically over South East Asia before and after the Indian monsoon. In our modelling approach, 90 vertical layers were used extending to the lower mesosphere. As discussed earlier, many studies showed that EMAC represent the vertical transport well and is used for many studies focusing on stratospheric processes.

Page 11: useful graphic and table. Could the authors also add a column to their table relating to their own results? What is the implications of modelling different years of fire inventories when comparing the results in Table 1? I would also be consistent with significant figures throughout the table.

Table 1 already included results from this study, but the presentation was confusing. The table was updated such that it is clearer that the column 2 and 3 give results from this study, but using the emission factors of Koss et al., 2018 and Kumar et al., 2018, respectively. A discussion on the different years simulated was added, as related to an earlier comment.

Page 11, line 8: is the wrong reference used here? Should it be Kumar et al 2018?

Correct. The reference was updated accordingly.

Page 12, starting at line 5: I would move this section before the global modelling data to help with the flow of the manuscript.

The NCO section is now moved to a more logical place directly after the other theoretical kinetic calculations.

Page 12, line 11: describe RAPRNOx

We now describe that the RAPRENOx process involves introducing HNCO in a combustion system through injection of cyanuric acid.

Page 12, lines 21-22: unclear statement. What is being referred to as "total rate coefficients"?

We now indicate that we did not do all possible reaction channels, but that the expected dominance of H-abstraction thus implies that it is a good estimate for the total rate of reaction through all reaction channels.

Page 12, lines 28-29: why not be consistent with p6, line 5 with 298K?

These numbers are for combustion applications (300-3000K), where "room temperature" is not a very important temperature and multiples of 100K seem more common. We also used this for the Arrhenius expression for HNCO + OH at combustion temperatures.

Conclusions:
Page 13, lines 18-19: 5 Gg/y out of (insert number of total losses).

This information was added to the manuscript.

Page 13, line 19: first time ammonia is mentioned in the conclusion – the authors can refer to (Leslie et al., 2019) for a back of the envelop calculation on ammonia budget from HNCO. How was ammonia implemented into the model? This discussion is rather important for the fate of HNCO and I would encourage the authors to discuss these numbers in the text as well.

The information on the implemented ammonia aqueous-phase mechanism is updated in the model description. Ammonia gas-phase sinks are already in the standard model version used. Aqueous-phase chemistry of ammonia is limited to the acid-base equilibrium in cloud droplets. With the current model setup we neglected ammonia (ammonium nitrate) in aerosols. We estimate HNCO hydrolysis produces ~120 Tg/yr of ammonia, which, on a global basis, contributes little to the ammonia budget. Our estimate is a factor 5-6 lower than the upper limit estimated by Leslie et al. (2019). We changed the text accordingly in section 4.

Page 13, lines 28-30: this sentence appears to be out of place, and outside the scope of this work. Unless a connection with the authors' HNCO modeling can be made?

We have added several pieces of information throughout the manuscript regarding transport of HNCO to the UTLS and the stratosphere, also related to the Indian Monsoon. As this part of the atmosphere is not the focus of the work at this time, we choose not to extend this aspect by more than the current additions, as this would require additional computational studies. The current work is related to ongoing work at our institute on HNCO emissions both at surface level and its role in the stratosphere.

Technical comments:
Reference of SUVA should have the acronym written out.

The English translation of the acronyms is now spelled out.

Page 1, line 28: what is meant by "first recognized"? First studied? First synthesis? I would encourage the authors to be more precise.

We now state explicitly that it's the molecular structure and synthesis that was discovered.

Page 2, line 21: instead of "slippage" did the authors means "seepage" (especially for gas seeping out)?

The text was rephrased.

Page 2, line 36: remove the word "very"

The text was rephrased.

Page 3, line 5: best to remove the word "recent".

Removed

Strangely enough, the numbers reported in the text do not match the numbers in the figure. Could the authors double check the numbers on: Page 5, line 24; page 5, line 26 (2 numbers); page 7, line 12.

The numbers match, but are rounded to the nearest kcal/mol in the text, as the extra digits serve no need and only makes the reading harder. To avoid confusion we added the additional digits to the text, matching the figure.

Page 9, line 23: should be written HNCO (not HCNO).

Corrected

Page 13, line 15: remove the word "fairly".

Removed

---

## Author Response (AR2)

**Reply to comments**

**Editor Decision: Publish subject to minor revisions (review by editor)** (01 May 2020) by James Roberts
Thanks for attending to the reviewer's comment. I have just a few more things that need your attention.

Abstract – Page 13, Line 16. It is a bit of an overstatement to say "known to be harmful" It would be better to
say 'could be' or 'suspected to be'. I think you explained it reasonably well later on in the paper.

Updated to "suspected"

Abstract – Page 13, Line 34. Eliminate the word "for" here.

Deleted

Abstract – Page 13, Line 38. Should say "rarely exceed levels above 10 pptv in other areas of the troposphere"

Corrected

Page 16, Line 13. I think you should keep the Barth et al., reference here, because Roberts and Liu use that
estimate for their fastest heterogeneous loss estimate (low pH clouds).

We agree, and reverted back to Barth et al.

Page 18, lines 22-23. Part of this sentence "not being considered by mean of a large and constant resistance"
doesn't make sense. Do you mean "not being considered by using a large and constant resistance" ?

The text has been rewritten as: "The submodel DDEP is used to simulate the dry deposition of HNCO using the
default scheme, where non-stomatal uptake is effectively disabled by using a large and constant resistance".

Page 23, Line 7. It looks like the rate constant for NCO + O2 has been measured:
https://kinetics.nist.gov/kinetics/Detail?id=1974SCH/SCH363:4 although I have not looked at the primary
source.

We thank the editor for finding this reference, as we were not aware of that paper. Only after extensive searches
were we able to find the page numbers, and even then, we were not able to find a library that carries this
publication and could provide us with a copy within a reasonable time span and at a cost commensurate to the
importance of this value in the manuscript. The rate coefficient is about what would be estimated for the
resonance-stabilized N.C=O radical, but having a measured rate coefficient in the paper would be useful for
readers less experienced in estimating such rate coefficients. We compromised by providing the measured rate
coefficient in the paper, but referring both to the original publication and the NIST database wherefrom we
sourced the value.

Page 24, Line 17. I suggest restructuring this to read "are negligible on a global scale".

Done

Page 24, Lines 27 and elsewhere through the presentation of the results. I don't think the uncertainties in the
model estimates justify your use of 6 significant figures here. It seems like 3 would be the most appropriate
number (e.g. 2519.61 should be 2520). This is true for other places you have presented results, really 3 sig. fig.
are the most that can be justified.

We only gave higher numbers of significant digits in the discussion section and Table 1; the conclusion already
used rounded values. We have now rounded all these numbers to 3 or even 2 significant digits throughout, to
better reflect the expected accuracy of the values.

ADDITIONAL CHANGES:
In anticipation of the typesetting of the manuscript, we improved the formatting of the manuscript in some
places, e.g. italicizing k and T, standardizing on spaces around units, etc. A few references were likewise
reformatted to better adhere to the ACP/ACPD style (not change-tracked due to bibliographic software).

[revised manuscript text omitted]